# Ice-Crystal Nucleation in Water: Thermodynamic Driving Force and Surface Tension. Part I: Theoretical Foundation

**DOI:** 10.3390/e22010050

**Published:** 2019-12-30

**Authors:** Olaf Hellmuth, Jürn W. P. Schmelzer, Rainer Feistel

**Affiliations:** 1Leibniz Institute for Tropospheric Research (TROPOS), Permoserstraße 15, D-04318 Leipzig, Germany; 2Institute of Physics, University of Rostock, Albert-Einstein-Straße 23-25, D-18059 Rostock, Germany; juern-w.schmelzer@uni-rostock.de; 3Leibniz Institute for Baltic Research (IOW), Seestraße 15, D-18119 Rostock-Warnemünde, Germany; rainer.feistel@io-warnemuende.de

**Keywords:** classical nucleation theory, crystallization thermodynamics, homogeneous freezing, thermodynamic driving force of nucleation, ice–water surface tension, Kauzmann temperature and pressure, TEOS-10

## Abstract

A recently developed thermodynamic theory for the determination of the driving force of crystallization and the crystal–melt surface tension is applied to the ice-water system employing the new Thermodynamic Equation of Seawater TEOS-10. The deviations of approximative formulations of the driving force and the surface tension from the exact reference properties are quantified, showing that the proposed simplifications are applicable for low to moderate undercooling and pressure differences to the respective equilibrium state of water. The TEOS-10-based predictions of the ice crystallization rate revealed pressure-induced deceleration of ice nucleation with an increasing pressure, and acceleration of ice nucleation by pressure decrease. This result is in, at least, qualitative agreement with laboratory experiments and computer simulations. Both the temperature and pressure dependencies of the ice-water surface tension were found to be in line with the le Chatelier–Braun principle, in that the surface tension decreases upon increasing degree of metastability of water (by decreasing temperature and pressure), which favors nucleation to move the system back to a stable state. The reason for this behavior is discussed. Finally, the Kauzmann temperature of the ice-water system was found to amount $T_{K}=116\;K$, which is far below the temperature of homogeneous freezing. The Kauzmann pressure was found to amount to $p_{K}=-212\;\mathrm{MPa}$, suggesting favor of homogeneous freezing on exerting a negative pressure on the liquid. In terms of thermodynamic properties entering the theory, the reason for the negative Kauzmann pressure is the higher mass density of water in comparison to ice at the melting point.

## 1. Introduction

### 1.1. Motivation

The outstanding importance of homogeneous freezing for a variety of natural and technical processes such as the microphysical evolution of atmospheric clouds (e.g., Meyers et al. [1], Khvorostyanov and Sassen [2], Lohmann and Kärcher [3], Lohmann et al. [4], Pruppacher and Klett [5], Heymsfield et al. [6], Jensen and Ackerman [7], Barahona and Nenes [8], Jensen et al. [9], Zasetsky et al. [10], Khvorostyanov and Curry [11], Khvorostyanov and Curry [12], Hellmuth et al. [13], Khvorostyanov and Curry [14], Lohmann et al. [15]), the cryopreservation of organelles, cells, tissues, extracellular matrices, organs, and foods (e.g., Pegg [16], Espinosa et al. [17], Espinosa et al. [18], see also https://en.wikipedia.org/wiki/Cryopreservation, visited on December 29, 2019), and water vitrification (e.g., Debenedetti and Stanley [19], Bhat et al. [20], Zobrist et al. [21]) stimulated a highly visible number of investigations on the thermophysical behavior of undercooled and deeply undercooled water within the framework of laboratory studies and evaluation of experimental data (e.g., McDonald [22], Butorin and Skripov [23], Hagen et al. [24], Hare and Sorensen [25], Henderson and Speedy [26], Speedy [27], Bartell and Huang [28], Gránásy [29], Huang and Bartell [30], Jeffery and Austin [31], Benz et al. [32], Holten et al. [33], Stöckel et al. [34], Souda [35], Tabazadeh et al. [36], Vortisch et al. [37], Malila and Laaksonen [38], Atkinson et al. [39]),by computer simulations (e.g., Gránásy [29,40], Matsumoto et al. [41], Oxtoby [42], Nada et al. [43], Laird and Davidchack [44], Vega and Abascal [45], Bai and Li [46], Bartell and Wu [47], Hernández de la Peña and Kusalik [48], Vega et al. [49], Vrbka and Jungwirth [50], Espinosa et al. [17,18], Moore and Molinero [51], Tanaka and Kimura [52]),and in form of fundamental theoretical considerations and synoptical views (e.g., Debenedetti and Stanley [19], Bartell [53], Ford [54], Debenedetti [55]).

Comprehensive overviews on the fundamental thermodynamic and molecular properties of water and the transition from clusters to liquid are given, e.g., by Ludwig [56], on undercooled and glassy water by Debenedetti [55], and on the notions, methods, and challenges to determine the crystal–melt interfacial free energy by Gránásy [29] and Laird and Davidchack [44]. Basic studies on the thermodynamic behavior of metastable liquids were performed, e.g., by Skripov [57], Skripov and Baidakov [58], Skripov and Koverda [59], Debenedetti et al. [60], Baidakov [61], Baidakov [62], Baidakov [63,64], Baidakov and Protsenko [65,66], Skripov and Faizullin [67], Baidakov et al. [68], Bartell and Wu [69]. In the last decade highly accurate equations of state (EoS) for water and ice became available, which are based on data from the experimentally accessible parts of the phase diagram of water: (i) for stable water (Wagner and Pruß [70], Wagner et al. [71], Guder [72]); (ii) for seawater (Feistel and Hagen [73], Feistel [74,75], Feistel et al. [76]) (iii) for hexagonal ice (Feistel and Hagen [77,78], Feistel and Wagner [79,80,81,82], IAPWS R10-06 [83]), (iii) for undercooled water (Holten et al. [84,85,86]). The application of these EoS’ is supported by the availability of international guidelines and standards for execution (Feistel et al. [87], Wright et al. [88], Feistel [89,90], IAPWS R6-95 [91], IAPWS R10-06 [83], IAPWS [92], IAPWS R13-08 [93], IAPWS [94,95], IAPWS G12-15 [96], IOC, SCOR, and IAPSO [97]). The aforementioned list of works contributing to water-to-ice crystallization, however, must inevitably remain incomplete and can be further extended.

The classical theory of nucleation (CNT) and growth processes is till now the major tool in the interpretation of experimental data on crystal nucleation and growth (e.g., Skripov [57], Skripov and Koverda [59], Skripov and Faizullin [67], Gutzow and Schmelzer [98], Gutzow and Schmelzer [99], Debenedetti [100], Kelton and Greer [101], Herlach et al. [102]). In its physical ingredients it is based on the thermodynamic theory of heterogeneous systems as developed by Josiah W. Gibbs (Gibbs [103,104]). Following Gibbs’ method in the specification of the properties of the critical clusters, it turns out that they correspond widely to the properties of the newly evolving macroscopic phases. This consequence of Gibbs’ theory gives the foundation of one of the main approximations of CNT in application to crystal nucleation, namely the identification of the bulk properties of the critical crystallites with the properties of the evolving macroscopic crystalline phase (Schmelzer and Abyzov [105]).

In line with such approximation, the surface tension in between melt and critical crystal can be identified with the respective value for a planar equilibrium coexistence of the respective liquid and crystalline phases. The latter assumption is denoted commonly as capillarity approximation. In the framework of CNT, frequently a curvature dependence of the surface tension is introduced in order to reconcile theory with experiment while the bulk properties of the critical clusters are assumed to be more or less defined as described above. Moreover, the introduction of a curvature dependence of the surface tension is the major tool to arrive at a correct description of nucleation rates measured experimentally. Alternatively, the theoretical expressions for the kinetic pre-factor in the expression for the steady-state nucleation rate can be modified. However, this approach results as a rule only in minor changes of the theoretical predictions (Gutzow and Schmelzer [98], Gutzow and Schmelzer [99], Skripov and Koverda [59]).

Alternative approaches have been advanced in recent decades based on generalizations of the classical Gibbs’ approach going beyond these simplest approximations (Gutzow and Schmelzer [99], Schmelzer et al. [106], Schmelzer and Abyzov [107]). These methods allow one to describe and in this way also to account for variations of the bulk properties of critical clusters in dependence on the degree of deviation from equilibrium. They are, however, much more complex and not as easily applicable as classical theory. Consequently, at least as a first estimate, CNT based on Gibbs’ classical method of description will also be retained in future to serve as a valuable tool in treating experimental data.

### 1.2. Rationale of the Present Study

Based on such considerations, in recent papers by Schmelzer and Abyzov [105,108] and Schmelzer et al. [109,110] two of the basic ingredients of CNT were revisited: the methods of specification of the thermodynamic driving force of nucleation and the dependence of the surface tension on the degree of deviation from equilibrium (i.e., the degree of metastability) or, equivalently, on the size of the critical clusters (Schmelzer et al. [111,112]). This analysis was performed for crystal nucleation caused by both variations of temperature and pressure. In particular, it was shown there that for both cases, the Tolman equation can be employed as an appropriate approximation for the description of the curvature dependence of the surface tension and not only for variations of external pressure at isothermal conditions as studied by Tolman [113]. Moreover, also going beyond Tolman’s analysis it is shown that Tolman’s approach can be employed also for multi-component systems provided the composition of the crystal phase (as employed as the basic assumption in CNT) and the composition of the liquid (as it is most frequently studied in crystallization) are considered as or kept constant. Consequences from the basic equations derived were discussed in the cited papers mainly for the most frequently occurring situation that the mass-specific volume of the crystal phase is smaller as compared to the respective value of the liquid phase.

Here, we will discuss the following aspects of ice nucleation in water as a very important example in many respects, wherein the opposite condition is fulfilled, i.e., where the mass-specific volume of the crystal phase is larger as compared to the respective value for the liquid phase:(i)As the first topic of the analysis, we will explore which qualitative differences arise in the water–ice system in comparison to other systems discussed earlier. Since we restrict the analysis here to a one-component case, it is also reasonable to expect that the basic assumptions of CNT may be fulfilled in a good approximation. At least, such conclusion was drawn quite recently based on molecular dynamics studies of melt crystallization for Lennard–Jones systems (Baidakov [64]). Possible generalizations of the theory in terms of the generalized Gibbs’ approach accounting for variations of density of the critical crystallites (as performed by some of us for the description of condensation and boiling (Schmelzer and Schmelzer Jr. [114,115], Schmelzer and Baidakov [116]), or segregation in solutions (Schmelzer et al. [117], Abyzov and Schmelzer [118], Schmelzer and Abyzov [119])) will not be discussed here. Having in mind the aforementioned importance of ice–crystal nucleation in a variety of processes in nature, we will further analyze in detail the degree of quantitative accuracy in the application of the general relations, derived in the mentioned papers, to this particular realization of crystal nucleation. This concerns the expressions for the temperature and pressure dependencies of the thermodynamic driving force of crystallization and the ice–water surface tension.(ii)A second aspect of the analysis is the application of the advanced Thermodynamic Equation of Seawater TEOS-10 for the calculation of the thermodynamic properties, which determine the temperature and pressure dependencies of the thermodynamic driving force of crystallization and the ice–water surface tension.(iii)Finally, the third question of interest is related to a problem already studied by Kauzmann [120], who asked for the behavior of a liquid at very low temperatures if experimental measurements were conducted sufficiently slowly to avoid glass formation, i.e., upon cooling the system slowly enough to ensure thermodynamic equilibrium of the liquid structure with the surroundings, but fast enough to avoid homogeneous crystallization. Kauzmann expected information about this behavior from the extrapolation of the known properties of supercooled liquids at temperatures above their glass-transformation temperature to low temperatures. The application of such extrapolation to plots of observed entropy vs. temperature for several substances (moist strikingly for glucose and lactic acid) revealed that below the glass-transition temperature (but still far above absolute zero) the extrapolated entropy of the liquid becomes less than that of the crystalline solid (Kauzmann [120] (Figure 4)). Similar tendencies were found for the extrapolated enthalpy vs. temperature and the specific volume vs. temperature, i.e., the liquid exhibits a tendency for lower enthalpy and smaller specific volume than the crystal well above absolute zero (Kauzmann [120], (Figures 3 and 6)). Such extrapolated behavior, however, is in contradiction with the lower entropy of the crystalline phase (in comparison with the liquid), which is expectable from the higher structural order of the crystalline phase. This contradiction became known as the Kauzmann paradox. Kauzmann concluded that the proposed extrapolation is not applicable. In order to solve this paradox, Kauzmann [120] argued that there must be a phase transition before the entropy of the liquid decreases. The exact wording of Kauzmann [120] (p. 224) reads: *“This peculiar result can only mean that somehow the above ’reasonable’ extrapolation is not permissible. The following resolution of the paradox is proposed: There is reason to believe that as the temperature is lowered the ’ambigious’ regions of the phase space intermediate between the definitely crystalline and definitely liquid regions begin to be able to contribute significantly to the partition function of the liquid. This means that the free energy barriers between the liquid and the crystal tend to become relatively small at low temperatures. In particular, the barrier to crystal nucleus formation, which tends to be very large just below the melting point, may at low temperatures be reduced to approximately the same height as the free energy barriers which impede molecular reorientations in the liquid and which were shown to be responsible for glass formation. Under these circumstances crystal nuclei will form and grow at about the same rate as the liquid changes its structure following a change in temperature or pressure. In other words, the time required for the liquid to crystallize becomes of the same order as the time required for it to change its structure following some change in its surroundings. If, then, measurements are to be made on such a liquid before it has had a chance to crystallize, these measurements must also be made before the liquid can bring its structure into equilibrium with its surroundings. But this means, as we have seen, that the liquid will behave as a glass. Thus, as the temperature of a liquid is lowered one is ultimately forced to study it as a glass if one wishes to study it as a liquid at all. A non-vitreous stable liquid cannot exist below a certain temperature, and it is operationally meaningless to extrapolate the entropy, energy, and specific volume curves below that temperature, as we tried to do with such peculiar results.”* The temperature, at which the entropies of the liquid and the crystal are equal are called Kauzmann temperature. In a similar way, Schmelzer et al. [109] defined the Kauzmann pressure. One aim of the present study is the determination of the Kauzmann temperature and Kauzmann pressure for the ice–water system.

The aforementioned investigations represent preparative steps for a future, more general description of ice crystallization in undercooled water within the framework of the generalized Gibbs approach. Such generalization aims at the removal of an essential restriction of the classical Gibbs theory, namely the assumed identity of the volume-specific properties of the newly evolving cluster phase with those of the corresponding macrophase.

The paper consists of two parts. Part I presented here is devoted to two issues: (i) the formulation of the theory underlying the determination of the temperature and pressure dependencies of the thermodynamic driving force of ice crystallization and the ice–water surface tension; and (ii) the numerical evaluation of the differences between the exact expressions (serving as reference) and various approximations of the theory. The subsequent Part II comprises a detailed intercomparison of the theory with previously published expressions of the thermodynamic driving force of ice crystallization and the ice–water surface tension based on laboratory experiments and computer simulations. Furthermore, the predicted nucleation rates will be compared with measured nucleation rates in order to assess the predictive power of the theoretical approach presented here.

The present Part I is structured as follows. In Section 2, the basic thermodynamic relations are described: (i) the dependence of the thermodynamic driving force on temperature and pressure; (ii) the dependence of the surface tension on temperature and pressure inclusive the parameters determining the curvature dependence of the surface tension of critical clusters; (iii) the equations for the determination of the Kauzmann temperature and pressure (Schmelzer et al. [106], Schmelzer et al. [109], Schmelzer et al. [110], Kauzmann [120]). The relations given in Section 2 are applied to ice–crystal nucleation in undercooled water. The required thermodynamic bulk properties of liquid and crystal phases of water are taken from the advanced EoS of seawater TEOS-10 (Feistel et al. [87] (Part 1); Wright et al. [88] (Part 2); IOC, SCOR, and IAPSO [97]; Feistel [89,90]), presented in Section 3. The results and discussion in Section 4 will complete the paper.

The three Appendices at the end of the paper include the derivation of the thermodynamic calculus applied here (Appendix A), the rationale of Kauzmann’s perception of metastability of undercooled liquids (Appendix B), and the details of the determination of the ice-water activation energy applied here in the nucleation rate calculus, respectively (Appendix C).

## 2. Basic Equations

### 2.1. Steady-State Nucleation Rate according to CNT

According to CNT, the steady-state rate, *J*, of homogeneous nucleation of critical clusters of phase α from its metastable maternal phase β reads (e.g., Pruppacher and Klett [5], Gutzow and Schmelzer [99], Hellmuth et al. [13]) (see Section A.1 and Section A.2):(1)$$\begin{array}{ccc} J & = & J_{\mathrm{kin}}\exp\left(-\frac{\Delta G_{c}^{\left(\mathrm{cluster}\right)}}{k_{B}T}\right)\;,\;\\ \Delta G_{c}^{\left(\mathrm{cluster}\right)} & = & \frac{1}{3}A_{\alpha }\sigma_{\alpha \beta }=\frac{16\pi }{3}\frac{\sigma_{\alpha \beta }^{3}}{{\left(\Delta g_{\mathrm{df},c}^{\left(\mathrm{bulk}\right)}\right)}^{2}}\;,\;\Delta g_{\mathrm{df},c}^{\left(\mathrm{bulk}\right)}=p_{\alpha }-p_{\beta }.\;\\ R_{\alpha } & = & \frac{2\sigma_{\alpha \beta }}{\Delta g_{\mathrm{df},c}^{\left(\mathrm{bulk}\right)}}. \end{array}$$

In Equation (Equation 1) the quantity $J_{\mathrm{kin}}$ is a kinetic prefactor determining the rate of cluster formation in the absence of a thermodynamic energy barrier. The latter is described by the Boltzmann term on the right-hand side of Equation (Equation 1) with $\Delta G_{c}^{\left(\mathrm{cluster}\right)}$ denoting the Gibbs free energy required to form a critical cluster (subscript c) with radius $R_{\alpha }$, surface area $A_{\alpha }=4\pi R_{\alpha }^{2}$, and surface tension $\sigma_{\alpha \beta }$. The physical quantity $k_{B}$ is the Boltzmann constant. The quantity $\Delta g_{\mathrm{df},c}^{\left(\mathrm{bulk}\right)}$ is called thermodynamic driving force of nucleation. It is determined originally by the pressure difference, $p_{\alpha }-p_{\beta }$, between the critical cluster of phase α and the maternal phase β.

However, in application to crystal nucleation alternative approaches for its specification are required and employed respectively. We will discuss them in Section 2.2. Please note that in the present approach, we consider critical crystal clusters as to be of spherical shape and employ the Gibbs’ treatment developed originally for fluid-like systems. The theoretical foundation of such treatment is discussed in detail in Schmelzer et al. [111,112].

### 2.2. Different Ways to Determine the Temperature and Pressure Dependence of the Thermodynamic Driving Force

#### 2.2.1. Exact form of the Thermodynamic Driving Force

According to Gibbs’ classical approach, the critical cluster of phase α is assumed to be in thermodynamic equilibrium with its maternal phase β, comprising mechanical equilibrium (Laplace equation), chemical (or diffusion) equilibrium, and thermal equilibrium between the coexisting macrophases α and β. For a one-component system these equilibrium conditions read (see Section A.2.1):(2)$$p_{\alpha }-p_{\beta }=\frac{2\sigma_{\alpha \beta }}{R_{\alpha }}\;,$$
(3)$${\hat{\mu }}_{\beta }\left(p_{\beta },T_{\beta }\right)-{\hat{\mu }}_{\alpha }\left(p_{\alpha },T_{\alpha }\right)=0\;,$$
(4)$$T_{\beta }-T_{\alpha }=0.$$

Here, ${\hat{\mu }}_{\alpha }$ and ${\hat{\mu }}_{\beta }$ are the mass-specific (indicated by the “wide hat” symbol $\hat{\left(\;\;\right)}$) chemical potentials of the respective macrophases α and β. Adopting the closure conditions $p_{\beta }=p$ and $T_{\beta }=T$, assuming that pressure and temperature in the ambient phase are given, and having at one’s disposal the knowledge of the chemical potentials of the considered component in both macrophases, the chemical equilibrium given by Equation (Equation 3) provides a condition for the direct determination of $p_{\alpha }=p_{\alpha }\left(p,T\right)$ and therewith for the thermodynamic driving force of nucleation, $\Delta g_{\mathrm{df},c}^{\left(\mathrm{bulk}\right)}$ according to Equation (Equation 1).

#### 2.2.2. Approximative Form of the Thermodynamic Driving Force

Alternatively, the thermodynamic driving force can be approximated as follows (Gutzow and Schmelzer [98], Gutzow and Schmelzer [99], Schmelzer and Abyzov [105], Schmelzer et al. [109], Schmelzer et al. [111]) (see Section A.2.2):(5)$${\left.\Delta g_{\mathrm{df},c}^{\left(\mathrm{bulk}\right)}\left(T,p\right)\right|}_{\mathrm{approx}}\approx {\hat{\rho }}_{\alpha }\left(p,T\right)\left[{\hat{\mu }}_{\beta }\left(p,T\right)-{\hat{\mu }}_{\alpha }\left(p,T\right)\right].$$

Here, ${\hat{\rho }}_{\alpha }\left(p,T\right)$ denotes the mass density of cluster phase α. 

#### 2.2.3. Thermodynamic Driving Force from the Gibbs Fundamental Equation

Equivalently, $\Delta g_{\mathrm{df},c}^{\left(\mathrm{bulk}\right)}\left(T,p\right)$ can also be determined from the governing equation for the total differential of the Gibbs free energy, *G*, of a homogeneous, single-component system of *n* molecules, entropy *S* and volume *V*, applied to the macrophases α and β (Schmelzer et al. [109], Equations (4)–(9)) (see Section A.2.3):(6)$$\begin{array}{ccc} {\left.\Delta g_{\mathrm{df},c}^{\left(\mathrm{bulk}\right)}\left(T,p\right)\right|}_{\mathrm{num}}\; & = & -\int_{T_{m}^{⋆}}^{T}\Delta s\left(T,p_{m}^{⋆}\right)\;dT+\int_{p_{m}^{⋆}}^{p}\Delta v\left(T,p\right)\;dp.\\ \Delta s(T,p) & = & \frac{{\hat{S}}_{\beta }\left(T,p\right)-{\hat{S}}_{\alpha }\left(T,p\right)}{{\hat{V}}_{\alpha }\left(T,p\right)}=\frac{\Delta \hat{S}\left(T,p\right)}{{\hat{V}}_{\alpha }\left(T,p\right)}\;,\;\\ \Delta v(T,p) & = & \frac{{\hat{V}}_{\beta }\left(T,p\right)-{\hat{V}}_{\alpha }\left(T,p\right)}{{\hat{V}}_{\alpha }\left(T,p\right)}=\frac{\Delta \hat{V}\left(T,p\right)}{{\hat{V}}_{\alpha }\left(T,p\right)}. \end{array}$$

Here, ${\hat{S}}_{\alpha ,\beta }$ and ${\hat{V}}_{\alpha ,\beta }$ denote the mass-specific entropies and mass-specific volumes of the respective macrophases α and β. The integration in Equation (Equation 6) starts at some particular α−β equilibrium state $\left(T_{m}^{⋆},p_{m}^{⋆}\right)$ (subscript *m*) and ends at an actual non-equilibrium state (T,p). The reference equilibrium state is set to $p_{m}^{⋆}=10^{5}\;\mathrm{Pa}$ and $T_{m}^{⋆}=273.15\;K$. The superscript ⋆ is used to distinguish the chosen reference state from any other equilibrium state along the melting line $\left(T_{m},p_{m}\right)$ with $T_{m}\left(p\right)$ denoting the melting temperature and $p_{m}\left(T\right)$ the melting pressure, respectively. The system is first transferred in a reversible isobaric process at $p=p_{m}^{⋆}$ from $T_{m}^{⋆}$ to *T*, and then subsequently transferred in an isothermal process at T=const. from $p_{m}^{⋆}$ to *p*, i.e., via the path $\left(T_{m}^{⋆},p_{m}^{⋆}\right)$→$\left(T,p_{m}^{⋆}\right)$→(T,p). As the Gibbs free energy is a thermodynamic potential, the difference in the mass-specific Gibbs free energy does not depend on the particular way to transfer the system from its equilibrium state $\left(T_{m}^{⋆},p_{m}^{⋆}\right)$ to any non-equilibrium state (T,p). Knowing ${\hat{S}}_{\alpha ,\beta }$ and ${\hat{V}}_{\alpha ,\beta }$, the driving force ${\left.\Delta g_{\mathrm{df},c}^{\left(\mathrm{bulk}\right)}\left(T,p\right)\right|}_{\mathrm{num}}$ can be obtained from Equation (Equation 6) by numerical integration.

#### 2.2.4. Linearized Form of the Thermodynamic Driving Force from the Gibbs Fundamental Equation

Expanding the integrands Δs(T,p) and Δv(T,p) in Equation (Equation 6) into Taylor series up to the linear terms, Schmelzer et al. [109] (Equation (Equation 23)) obtained the following analytical solution of the integral, Equation (Equation 6) (see Section A.2.4):(7)$$\begin{array}{c} {\left.\Delta g_{\mathrm{df},c}^{\left(\mathrm{bulk}\right)}\left(T,p\right)\right|}_{\mathrm{lin}}\approx \Delta h_{m}\frac{\Delta T}{T_{m}^{⋆}}\left(1-\gamma_{T,m}\frac{\Delta T}{2T_{m}^{⋆}}\right)+\Delta v_{m}\Delta p\left(1-\gamma_{p,m}\frac{\Delta p}{2p_{m}^{⋆}}\right)\;,\;\\ \gamma_{T,m}=\frac{\Delta {\hat{c}}_{p,m}}{\Delta {\hat{S}}_{m}}\;,\;\gamma_{p,m}=\frac{p_{m}^{⋆}\Delta \kappa_{T,m}}{\epsilon_{m}\Delta v_{m}}. \end{array}$$

Here, $\Delta T=T_{m}^{⋆}-T$ is the temperature difference, called undercooling for $T<T_{m}^{⋆}$. Analogously, $\Delta p=p-p_{m}^{⋆}$ is the pressure difference, corresponding to an overpressure for $p>p_{m}^{⋆}$ and to an underpressure for $p<p_{m}^{⋆}$. The quantity $\Delta h_{m}=\Delta {\hat{H}}_{M,m}/{\hat{V}}_{\alpha }\left(T_{m}^{⋆},p_{m}^{⋆}\right)$ is the volumetric melting enthalpy with $\Delta {\hat{H}}_{M,m}=\Delta {\hat{H}}_{M}\left(T_{m}^{⋆}\right)$ denoting the mass-specific enthalpy of melting at temperature $T_{m}^{⋆}$. Furthermore, $\Delta v_{m}=\Delta {\hat{V}}_{m}/{\hat{V}}_{\alpha }\left(T_{m}^{⋆},p_{m}^{⋆}\right)$, with $\Delta {\hat{V}}_{m}={\hat{V}}_{\beta }\left(T_{m}^{⋆},p_{m}^{⋆}\right)-{\hat{V}}_{\alpha }\left(T_{m}^{⋆},p_{m}^{⋆}\right)$ denoting the difference of the mass-specific volumes, $\Delta {\hat{c}}_{p,m}={\hat{c}}_{p,\beta }\left(T_{m}^{⋆},p_{m}^{⋆}\right)-{\hat{c}}_{p,\alpha }\left(T_{m}^{⋆},p_{m}^{⋆}\right)$ the difference of the mass-specific isobaric heat capacities, $\Delta {\hat{S}}_{m}={\hat{S}}_{\beta }\left(T_{m}^{⋆},p_{m}^{⋆}\right)-{\hat{S}}_{\alpha }\left(T_{m}^{⋆},p_{m}^{⋆}\right)$ the difference of the mass-specific entropies, $\Delta \kappa_{T,m}=\kappa_{T,\beta }\left(T_{m}^{⋆},p_{m}^{⋆}\right)-\kappa_{T,\alpha }\left(T_{m}^{⋆},p_{m}^{⋆}\right)$ the difference of the isothermal compressibilities between macrophases α and β, and $\epsilon_{m}={\hat{V}}_{\alpha }\left(T_{m}^{⋆},p_{m}^{⋆}\right)/{\hat{V}}_{\beta }\left(T_{m}^{⋆},p_{m}^{⋆}\right)$, respectively. In comparison with Equation (Equation 5), Equation (Equation 7) has the huge advantage that the driving force is expressed in terms of directly measurable thermodynamic parameters and of the deviations of temperature and pressure from the respective parameters of the chosen macroscopic equilibrium state. By this reason, not relations in the form of Equation (Equation 5), but in the form of Equation (Equation 7) are commonly employed in the theoretical analysis of crystal nucleation processes. A similar relation we will derive in the next section with respect to the surface tension.

### 2.3. Temperature and Pressure Dependence of the Ice–Water Surface Tension

The crystal–melt interface energy has a large impact on the thermodynamic energy barrier for homogeneous freezing, because it enters the expression of the critical formation work by the power to three, i.e., $\Delta G_{c}^{\left(\mathrm{cluster}\right)}\propto \sigma_{\alpha \beta }^{3}$. Nevertheless, *“This interface energy is almost never known in supercooled liquids”* (Vortisch et al. [37]). According to Bai and Li [46], interfacial energies are, unfortunately, very weak and extremely difficult to obtain experimentally for systems with two condensed phases such as solid–liquid systems. Consequently, much work has been devoted to the determination of the surface tension at the crystal–melt interface (e.g., McDonald [22], Bartell [53], Huang and Bartell [30], Gránásy [29], Gránásy [40], Jeffery and Austin [31], Laird and Davidchack [44], Bai and Li [46], Baidakov [63], Baidakov et al. [121], Espinosa et al. [17], Espinosa et al. [18], Ickes et al. [122]). According to Bartell [53] (pp. 1083–1084), the surface tension is argued to play a role analogously to that of the activation energy in the kinetics of chemical reactions. The author further wrote that although its name is suggestive of a thermodynamic variable, the surface tension is a kinetic parameter whose most important role is to facilitate the estimation of nucleation rates at greater or smaller degrees of undercooling from a given measured nucleation rate. To what extent $\sigma_{\alpha \beta }$ reflects the true thermodynamic variable in serving as a closure parameter to explain freezing experiments has not be determined very precisely so far. *Ibidem*, this originates from the obvious difficulties to measure the work required to increase the interfacial area between a solid and another phase without performing other work (e.g., elastic or plastic deformation). The possibility of the coexistence of two phases at equilibrium at ambient pressure at only a single temperature poses another problem. With reference to theoretical considerations, $\sigma_{\alpha \beta }$ might be considered to have a physical meaning only at that single temperature and not at the deep undercooling encountered in nucleation experiments. As CNT is argued to have only qualitative validity, Bartell [53] considered $\sigma_{\alpha \beta }$ to be to some extent *“a bit of a fiction”*. Similar problems were already discussed by Gibbs in connection with the problem down to which critical cluster sizes thermodynamic concepts are applicable.

A comprehensive evaluation of methods to determine the ice-water surface tension and its temperature dependence was performed by Ickes et al. [123] (Section 4.1). According to these authors, owing to sampling problems and the onset of heterogeneous freezing of undercooled water on parts of any experimental setup, direct measurements of $\sigma_{\alpha \beta }$ are restricted to macroscopic water drops at temperatures $T\geq T_{m}^{⋆}=273.15\;K$. These measurements are then extrapolated to ice crystals of microscopic sizes in undercooled water, either by fitting $\sigma_{\alpha \beta }$ to measured nucleation rates employing CNT (e.g., Jeffery and Austin [31]), or alternatively by theoretical considerations and molecular models (e.g., Espinosa et al. [17,18]).

According to Schmelzer and Abyzov [108], Schmelzer et al. [109] (Equation (Equation 30)), and Schmelzer et al. [110], the dependence of the surface tension of critical crystallites on pressure and temperature can be expressed for small deviations from equilibrium as
(8)$$\frac{\sigma_{\alpha \beta }\left(T,p\right)}{\sigma_{\alpha \beta ,m}}≅\frac{T\Delta S(T,p)}{T_{m}\Delta S_{m}}=\frac{T\Delta \hat{S}\left(T,p\right)}{T_{m}\Delta {\hat{S}}_{m}}.$$

Here, $\sigma_{\alpha \beta ,m}=\sigma_{\alpha \beta }\left(T_{m}^{⋆},p_{m}^{⋆}\right)$ denotes the surface tension at the melting point, and $\Delta \hat{S}\left(T,p\right)$ and $\Delta {\hat{S}}_{m}$ are defined in Equations (Equation 6) and (Equation 7). By linearization of the scaling law given by Equation (Equation 8), Schmelzer and Abyzov [108], Schmelzer et al. [109] (Equation (Equation 32)), and Schmelzer et al. [110] derived the following expression for the temperature and pressure dependence of the surface tension of critical crystallites (see Section A.3):(9)$$\begin{array}{ccc} \frac{\sigma_{\alpha \beta }\left(T,p\right)}{\sigma_{\alpha \beta ,m}} & ≅ & \frac{T}{T_{m}^{⋆}}\left(1-\gamma_{T,m}\frac{\Delta T}{T_{m}^{⋆}}-\chi_{p,m}\frac{\Delta p}{p_{m}^{⋆}}\right)\;,\;\chi_{p,m}=\frac{p_{m}^{⋆}\Delta \alpha_{p,m}}{\Delta s_{m}}. \end{array}$$

In Equation (Equation 9), the quantity $\Delta \alpha_{p,m}=\alpha_{p,\beta }$$\left(T_{m}^{⋆},p_{m}^{⋆}\right)-\alpha_{p,\alpha }\left(T_{m}^{⋆},p_{m}^{⋆}\right)$ denotes the difference of the isobaric thermal expansion coefficients between macrophases α and β, and $\Delta s_{m}=\Delta {\hat{S}}_{m}/{\hat{V}}_{\alpha }\left(T_{m}^{⋆},p_{m}^{⋆}\right)$.

According to Gibbs [103], the surface tension of a crystallite depends on its curvature. The shape of this dependence was elaborated by Tolman [113]. Generalizing Tolman’s formula, Schmelzer et al. [112] derived the following expression for the curvature dependence of the surface tension (Schmelzer et al. [111], Schmelzer et al. [112] (Equations (3), (33), (34) and references)):(10)$$\sigma_{\alpha \beta }\left(R_{\alpha }\right)=\frac{\sigma_{\alpha \beta ,\infty }}{1+\frac{2\delta (R_{\alpha })}{R_{\alpha }}}\;,\;\Delta \approx \delta_{\infty }\left(1+\frac{l_{\infty }^{2}}{2\delta_{\infty }R_{\alpha }}\right)\;,\;\sigma_{\alpha \beta ,\infty }=\sigma_{\alpha \beta ,m}.$$

Here, δ denotes the Tolman parameter. At low degree of metastability the curvature of the critical embryo is small and the Tolman parameter approaches its planar equilibrium value, $\delta =\delta_{\infty }$. The parameter $l_{\infty }$ is as further length scale to account for the higher-order contribution to the approximation of the curvature depence of the surface tension. For the case of constant pressure, $p=p_{m}^{⋆}$, and weak undercooling one arrives at the following expression for $\delta_{\infty }$ in the limit $T\to T_{m}^{⋆}$ (superscript (T)) (Schmelzer et al. [111] (Equation (Equation 69)) (see Section A.3):(11)$$\delta_{\infty }^{\left(T\right)}{|}_{p=p_{m}^{⋆}}\approx \frac{\sigma_{\alpha \beta ,m}}{\Delta h_{m}}\left(1+\gamma_{T,m}\right).$$

Analogously, for the case of constant temperature, $T=T_{m}^{⋆}$, and sufficiently weak deviations of the pressure from $p_{m}^{⋆}$ one obtains the following dependence of the Tolman parameter in the limit $p\to p_{m}^{⋆}$ (superscript (p)) (Schmelzer et al. [111]), (see Section A.3):(12)$$\delta_{\infty }^{\left(p\right)}{|}_{T=T_{m}^{⋆}}\approx \frac{\sigma_{\alpha \beta ,m}}{p_{m}^{⋆}\Delta v_{m}}\chi_{p,m}.$$

For later comparison (see Section 4.2) of the different derived expressions for the temperature and pressure dependence of the surface tension, $\sigma_{\alpha \beta }$, we take the expression proposed by Jeffery and Austin [31] (Equation (Equation 8)) as the reference surface tension, which is based on the Turnbull formula (Turnbull [124]) for $\sigma_{\alpha \beta }$, proposed for application to several metals and metalloids. By addition of a correction term, Jeffery and Austin [31] (Equation (Equation 8)) re-fitted the Turnbull expression to experimental data of homogeneous water-to-ice nucleation rates from chamber experiments at p=0.1MPa in combination with CNT application:(13)$$\begin{array}{ccc} \sigma_{\alpha \beta }\left(T,p\right) & = & \underset{\mathrm{Turnbull}}{\underset{︸}{\varkappa_{T}\Delta {\hat{H}}_{M}\left(T\right){\left[{\hat{\varrho }}_{\alpha }\left(T,p\right)\right]}^{2/3}{\left(\frac{M_{w}}{N_{A}}\right)}^{1/3}}}+\Delta \sigma_{\alpha \beta }\;,\;\\ \Delta \sigma_{\alpha \beta } & = & -\varkappa_{\sigma }T\;,\;\varkappa_{T}=0.32\;,\;\varkappa_{\sigma }=9\times 10^{-5}\;J\;m^{-2}\;K^{-1}. \end{array}$$

Here, $\Delta {\hat{H}}_{M}\left(T\right)$ and ${\hat{\varrho }}_{\alpha }\left(T,p\right)$ denote the previously introduced mass-specific melting enthalpy and mass density of ice, $M_{w}$ is the molar mass of water, and $N_{A}$ the Avogadro constant. The excess value $\Delta \sigma_{\alpha \beta }$ was introduced as an empirical correction term, which depends only on temperature (see Appendix B for discussion). The parameter setting of $\varkappa_{T}$ and $\varkappa_{\sigma }$ in the original paper of Jeffery and Austin [31] is based on the use of the EoS of water developed by Jeffery [125] in combination with a special formulation of the kinetic prefactor $J_{\mathrm{kin}}$. In contrast to this, in the present evaluation of Equation (Equation 13) the thermophysical parameters $\Delta {\hat{H}}_{M}\left(T\right)$ and ${\hat{\varrho }}_{\alpha }\left(T,p\right)$ were taken from TEOS-10. One can safely expect that the differences in the behavior of $\sigma_{\alpha \beta }\left(T,p\right)$ between Equation (Equation 13) and the expressions drived below are primarily caused by differences in the physical foundation of the respective expressions but not by differences in the employed EoS for water.

The ratio $\sigma_{\alpha \beta }\left(T,p\right)/\sigma_{\alpha \beta ,m}$ according to Equation (Equation 13) is presented as a function of ΔT and Δp in Table 1. The surface tension remarkably decreases with decreasing temperature (increasing undercooling) and decreasing pressure (or, equivalently, with increasing degree of metastability of the fluid). One should keep in mind, however, that the parameters in Equation (Equation 13) were adjusted to data at atmospheric pressure. Therefore, the data at Δp>0 represent, strictly speaking, extrapolations.

### 2.4. Kauzmann Temperature and Pressure

In his seminal paper Kauzmann [120] discussed in detail the possibility that the entropy differences between liquid and crystal may approach zero at low temperatures denoted today as Kauzmann temperature, $T_{K}$ (see Schmelzer et al. [110] and Schmelzer and Tropin [126] for a detailed discussion). According to Debenedetti et al. [60], $T_{K}$ imposes a sharply defined thermodynamic limit to the possible existence of the liquid state of a given substance, since upon further undercooling the hypothetical liquid would have a lower entropy than the corresponding crystalline phase (referred to as “entropy catastrophe”). *Ibidem* Debenedetti et al. [60] concluded that the Kauzmann temperature is unattainable because the slowing down of molecular motion inevitably drives kinetically controlled glass transitions.

As shown recently with respect to crystal nucleation, the Kauzmann temperature exhibits the interesting peculiarity that the thermodynamic driving force does assume a maximum there (Schmelzer and Abyzov [105], Schmelzer et al. [106]). Indeed, the fulfillment of the condition $\Delta s(T_{K},p_{m}^{⋆})=0$ in Equation (Equation 6) leads immediately to a maximum of $\Delta g_{\mathrm{df},⋆}^{\left(\mathrm{bulk}\right)}\left(T_{K},p_{m}^{⋆}\right)$.

In analogy to the Kauzmann temperature, Schmelzer and Abyzov [105] and Schmelzer et al. [109] introduced the concept of Kauzmann pressure, $p_{K}$, defined by the condition $\Delta v(T_{m}^{⋆},p_{K})=0$ in Equation (Equation 6), leading to a maximum of $\Delta g_{\mathrm{df},⋆}^{\left(\mathrm{bulk}\right)}\left(T_{m}^{⋆},p_{K}\right)$. The Kauzmann temperature and pressure are determined by the following expressions (Schmelzer et al. [109] (Equations (24) and (26))) (see Section A.4):(14)$$T_{K}=T_{m}^{⋆}\left[\frac{\gamma_{T,m}-1}{\gamma_{T,m}}\right]\;,\;p_{K}=p_{m}^{⋆}\left[\frac{\gamma_{p,m}+1}{\gamma_{p,m}}\right].$$

## 3. The Advanced Thermodynamic Equation of Seawater TEOS-10

The basic equations presented in Section 2 were previously applied to crystallization of glass-forming melts, e.g., by Schmelzer and Abyzov [105,107,108], Schmelzer et al. [109], Schmelzer et al. [106,110,111,112], and Schmelzer and Tropin [126]. In the present study, this calculus will be applied to ice-forming melts, i.e., to undercooled water (phase β) and hexagonal ice (phase α). The reqired thermodynamic data are taken from an advanced seawater standard, the International Thermodynamic Equation Of Seawater 2010 (TEOS-10), which was adopted in June 2009 by the International Oceanographic Commission of United Nations Educational, Scientific and Cultural Organisation (UNESCO/IOC) on its 25th General Assembly in Paris. TEOS-10 is–to the best of our knowledge–the most advanced thermodynamic standard for the thermodynamic properties of pure water, water vapor, hexagonal ice, and seawater. It is essentially based on IAPWS certificated equations of state, which is in the case of pure water the standard IAPWS-95. Owing to its reliance on a large amount of experimental data on water and the inclusion of a comprehensive uncertainty analysis, IAPWS-95 (as a part of TEOS-10) is the best thermodynamic formulation of pure water. The notion “EoS of seawater” for TEOS-10 is due to its origin to serve as a generalized standard to describe seawater, which includes pure water as a special case (zero salinity).

To support the application of TEOS-10, a comprehensive source code library for the thermodynamic properties of liquid water, water vapor, ice, seawater, and humid air, is available and referred to as the Sea–Ice–Air (SIA) library. The background information and equations (including references for the primary data sources) required for the determination of the properties of single phases and components as well as of phase transitions and composite systems as implemented in the library are presented in two key papers of Feistel et al. [87] (Part 1) and Wright et al. [88] (Part 2), in the TEOS-10 Manual (IOC, SCOR, and IAPSO [97]), in an introductory paper of Feistel [89] and a comprehensive review paper of Feistel [90].

TEOS-10 is based on four independent thermodynamic functions, which are defined in terms of the independent observables temperature, pressure, density, and salinity:a Helmholtz function of fluid water, known as IAPWS-95 (Wagner and Pruß [70], IAPWS R6-95 [91]),a Gibbs function of hexagonal ice (Feistel and Wagner [82], IAPWS R10-06 [83]),a Gibbs function of seasalt dissolved in water (Feistel [74,75], IAPWS R13-08 [93]), anda Helmholtz function for dry air (Lemmon et al. [127]).

In combination with air–water cross-virial coefficients (Hyland and Wexler [128], Harvey and Huang [129], Feistel et al. [130]) this set of thermodynamic potentials is used as the primary standard for pure water (in liquid, vapor, and solid states), seawater, and humid air from which all other properties are derived by mathematical operations, i.e., without the need for additional empirical functions.

The IAPWS-95 fluid water formulation is based on ITS-90 and on the evaluation of a comprehensive and consistent data set, which was assembled from a total of about 20,000 experimental data of water. The authors of this water standard took into account all available information given in the scientific articles describing the data collection and critically reexamined the available data sets regarding their internal consistency and their basic applicability for the development of a new equation of state for water. Only those data were incorporated into the final nonlinear fitting procedure, which were judged to be of high quality. These selected data sets took into account experimental data which were available by the middle of the year 1994 (Wagner and Pruß [70]). The availability of reliable experimental data on undercooled liquid water was restricted to a few data sets for several properties only along the isobar p=1013.25hPa (Wagner and Pruß [70] (Section 7.3.2)), which set the lower limit of the temperature range of IAPWS-95 (and so of TEOS-10) to T=236K$\left(\vartheta =-37.15{\;}^{\circ }C\right)$. This temperature is called the temperature of homogeneous ice nucleation (or homogeneous freezing temperature), $T_{H}$, which represents the lower limit below which it is very difficult to undercool water. The thermodynamic functions from the SIA source code library, which are used in the present analysis, are given in Table 2.

By virtue of the definition range of TEOS-10, its application to liquid water is restricted to temperatures $T\geq T_{H}$.

## 4. Results and Discussion

### 4.1. Thermodynamic Driving Force of Water-to-Ice Nucleation

Table 3 contains the key thermodynamic parameters of the ice-water system at the reference equilibrium state $\left(T_{m}^{⋆},p_{m}^{⋆}\right)$, which are used for the subsequent calculations.

In Table 4 the exact, TEOS-10-based thermodynamic driving force of the ice-water system, $\Delta g_{\mathrm{df},c}^{\left(\mathrm{bulk}\right)}=p_{\alpha }-p_{\beta }$ according to Equation (Equation 1) with $p_{\alpha }$ determined from numerical solution of the transcendental Equation (Equation 3) (chemical equilibrium), is presented as a function of undercooling $\Delta T=T_{m}^{⋆}-T$ and the pressure difference $\Delta p=p-p_{m}^{⋆}$.

Negative values of $\Delta g_{\mathrm{df},c}^{\left(\mathrm{bulk}\right)}$ mean that there is no driving force to nucleation, i.e., the formation of ice crystallites from water, which is not undercooled any longer then, is impossible. The driving force to ice nucleation (or equivalently, the degree of metastability of the fluid) increases upon increasing undercooling and decreasing pressure, i.e., starting at $p_{m}^{⋆}$, a pressure difference $\Delta p=p-p_{m}^{⋆}<0$ favors crystallization of water, Δp>0 disfavors it.

The relative deviations (in percent) of the approximative, the numerical, and the linearized thermodynamic driving forces ${\left.\Delta g_{\mathrm{df},c}^{\left(\mathrm{bulk}\right)}\right|}_{X}$, X={approx,num,lin} according to Equations (Equation 5)–(Equation 7) from the exact driving force, $\Delta g_{\mathrm{df},c}^{\left(\mathrm{bulk}\right)}$ according to Equation (Equation 1), are presented in Table 5, Table 6 and Table 7. The relative deviation of the approximation ${\left.\Delta g_{\mathrm{df},⋆}^{\left(\mathrm{bulk}\right)}\right|}_{\mathrm{approx}}$ from the exact value remains far below one percent throughout the considered ranges of undercooling and pressure difference. Also the numerical solution ${\left.\Delta g_{\mathrm{df},c}^{\left(\mathrm{bulk}\right)}\right|}_{\mathrm{num}}$ is still a very good representation of the driving force throughout the considered range of undercooling and from zero until moderate pressure difference (0MPa≤Δp≤10MPa). The maximum of the relative deviation was found to amount 7% at Δp=100MPa for ΔT=10K. The same proposition with respect to accuracy holds also for the performance of the linearized representation of the driving force given by ${\left.\Delta g_{\mathrm{df},c}^{\left(\mathrm{bulk}\right)}\right|}_{\mathrm{lin}}$, which is based on a higher degree of approximation. While the linearized form is still a very good approximation of the exact driving force (relative deviation <2%) throughout the considered range of undercooling and pressure differences in the interval 0MPa≤Δp≤10MPa, the relative deviation increases to a maximum of 50% at Δp=100MPa (for ΔT=10K), which originates from the linearization applied in the derivation of the driving force. At these conditions, however, the nucleation rate is already very small.

### 4.2. Temperature and Pressure Dependence of the Ice–Water Surface Tension

The relative deviations of the ratio $\sigma_{\alpha \beta }\left(T,p\right)/\sigma_{\alpha \beta ,m}$ according to Equations (Equation 8) and (Equation 9) (Schmelzer et al. [109] (Equations (30) and (32))) from the reference ratio given by Equation (Equation 13) (Jeffery and Austin [31] (Equation (Equation 8))) are presented in Table 8 and Table 9, respectively. Both equations show qualitatively the same dependencies on temperature and pressure as the Jeffery–Austin expression, but the absolute values are in both cases considerably smaller beginning at moderate undercooling (e.g., maximum deviation of −34% for Equation (Equation 8) at ΔT=39K and Δp=0). Equations (Equation 8) and (Equation 9) behave quite similarly, i.e., the linearization of Equation (Equation 8) does not cause a substantial loss of information in comparison to the nonlinear function for $\sigma_{\alpha \beta }\left(T,p\right)$ given by Equation (Equation 8).

Table 10 shows the temperature and pressure coefficients, $\partial \sigma_{\alpha \beta }/\partial T$ and $\partial \sigma_{\alpha \beta }/\partial p$, derived for the linearized form of $\sigma_{\alpha \beta }\left(T,p\right)$ (Equation (Equation 9)) as a function of ΔT and Δp:(15)$$\frac{\partial \sigma_{\alpha \beta }}{\partial T}=\frac{\sigma_{\alpha \beta }}{T}\left[1+\gamma_{T,m}\frac{\sigma_{\alpha \beta ,m}}{\sigma_{\alpha \beta }}{\left(\frac{T}{T_{m}^{⋆}}\right)}^{2}\right]\;,\;\frac{\partial \sigma_{\alpha \beta }}{\partial p}=-\chi_{p,m}\frac{\sigma_{\alpha \beta ,m}}{p_{m}^{⋆}}\left(\frac{T}{T_{m}^{⋆}}\right).$$

Here, $\sigma_{\alpha \beta ,m}=31.2\times 10^{-3}\;J\;m^{-2}$ was determined from Equation (Equation 13). In accordance with the temperature and pressure dependencies presented in Table 1, Table 8, and Table 9 both coefficients are positive definite, i.e., $\partial \sigma_{\alpha \beta }/\partial T>0$ and $\partial \sigma_{\alpha \beta }/\partial p>0$. A positive temperature coefficient of the surface tension was reported, e.g., for mercury, tin, and sodium by Skripov and Faizullin [67] (Equations (3.84), (3.85) and Figures 3.29, 3.30), for the Lennard–Jones system (a prototype model for the interactions of neutral nonpolar molecules) by Laird and Davidchack [44] (Table 2), Bai and Li [46] (Figure 12), and Baidakov [63] (Figures 1, 2, and Equation (Equation 3)) (Baidakov [63] reanalyzed and readjusted the scaling law proposed by Skripov and Faizullin [67] (Equations (3.84) and (3.85)) to bring the scaling-law predictions in agreement with his MD simulations), and for water by McDonald [22], Wood and Walton [131], Bartell [53] (Figure 6), Gránásy [29] (Figure 4), Gránásy [40] (Figure 7), Jeffery and Austin [31], and Tanaka and Kimura [52]. The positive temperature coefficient of the surface tension is argued to originate from the entropy loss in the liquid due to the ordering near the crystal–melt interface (e.g., Gránásy [29], Gránásy [40], Bai and Li [46] (see reference therein to Spaepen)).

According to Section 4.1, the driving force of nucleation as a measure of the degree of metastability of the fluid was found to increase upon decreasing temperature and decreasing pressure. The surface tension of the ice-water system responds to increasing metastability in such a way that the freezing probability increases to remove the metastability and to adjust the system back to equilibrium. Hence, the decrease of the surface tension with decreasing temperature and pressure is in agreement with the principle of le Chatelier–Braun (Landau and Lifschitz [132] (pp. 61–64)): variations of external parameters are expected to counteract the initial perturbation to bring the system back to equilibrium. The positive definiteness of $\partial \sigma_{\alpha \beta }/\partial p$ is caused by the parameter $\chi_{p,m}=-2\times 10^{-5}<0$ according to Equation (Equation 9) and Table 3, which, in turn, is caused by $\Delta \alpha_{p,m}=\alpha_{p,\beta }\left(T_{m}^{⋆},p_{m}^{⋆}\right)-\alpha_{p,\alpha }\left(T_{m}^{⋆},p_{m}^{⋆}\right)<0$ (Table 3), i.e., by the higher thermal expansion coefficient of ice as compared to water. Molecular-theoretical arguments for the described pressure dependence will be given below.

An analysis of a large sample of empirical, theoretical, and simulated $\sigma_{\alpha \beta }\left(T\right)$ correlations performed by Ickes et al. [123] (Figures 2 and 3, Table 3) revealed a large scatter of both the surface tension ($\sigma_{\alpha \beta }\left(273.15\;K\right)=\left(10-44\right)\times 10^{-3}\;J\;m^{-2}$ and $\sigma_{\alpha \beta }\left(220\;K\right)=\left(6.8-26.7\right)\times 10^{-3}\;J\;m^{-2}$) and its temperature coefficient $\left(\partial \sigma_{\alpha \beta }/\partial T=\left(0.1-0.25\right)\times 10^{-3}\;J\;m^{-2}\;K^{-1}\right)$. The temperature coefficient presented in Table 10 exhibits a weak decrease upon increasing undercooling with values located at the lower end of the range reported by Ickes et al. [123]. The experimental data of Bartell and Huang [28] (Figure 8) and the simulation data of Espinosa et al. [17] (Figure 4 and Table 2) and [18] (Figure 1d) fit also well into the ranges of $\sigma_{\alpha \beta }\left(T\right)$ and $\partial \sigma_{\alpha \beta }/\partial T$ reported by Ickes et al. [123]. In their freezing experiments on homogeneous water-to-ice nucleation Huang and Bartell [30] (Equation (Equation 3)) employed the following temperature dependence of the ice-water surface tension:(16)$$\frac{\sigma_{\alpha \beta }\left(T\right)}{\sigma_{\alpha \beta }\left(T_{0}\right)}={\left(\frac{T}{T_{0}}\right)}^{n}\;,\;n\approx 0.3.$$

Here, $T_{0}$ serves as a reference temperature. Based on experimental nucleation data at ≈242K and 200K, Bartell [53] (Figures 5 and 6) and Huang and Bartell [30] (Figures 7 and 8) reported the exponent to be in the range n=0.3−0.4. According Bartell [53] (Figure 6 and references), the values n=0.3−0.4 derived from his experimental approach refer to cubic ice. Extrapolation of the surface tension from the undercooled regime to T=273.15K according to $\sigma_{\alpha \beta }\propto T^{n}$ yields $\sigma_{\alpha \beta }\left(273\;K\right)\approx 24\;\mathrm{mJ}\;m^{-2}$, which is by $\approx 9\;\mathrm{mJ}\;m^{-2}$ lower than the value derived from equilibrium contact angles between water and two crystals of hexagonal ice sharing a grain boundary. Bartell noted that $75\;\overset{˚}{A}$ molecular clusters, cooled down to 200K (cubic ice) by evaporation, manage to avoid the extreme anomalies proposed to occur in bulk water in the vicinity of 226K if nucleation could be avoided. According to Huang and Bartell [30] (p. 3927, see references therein to Turnbull and Spaepen), the exponent *n* is expected to be positive rather than negative. The authors argued, that the free energy of the interface should increase as temperature rises as the interfacial entropy tends to be negative, because a liquid in contact with crystal is forced into a structure more ordered than that of the bulk.

Reanalyzing the temperature dependence in Equation (Equation 13) in the form given by Equation (Equation 16), one obtains n=1.63−2.85 (depending on temperature and pressure), and performing the same analysis for Equation (Equation 9), one arrives at n=1.82−2.73. Hence, the power *n* of the temperature dependence of the expressions analyzed in the present study is considerably larger than that used by Huang and Bartell [30]. Based on CNT and using MD simulations of a Lennard–Jones system to setup the nucleation scenario, Bai and Li [46] (Figure 12) derived a best-fit linear dependence of the solid–liquid surface tension on temperature, i.e., n=1, with a positive temperature coefficient. The tendency of the temperature dependence of the surface tension was reported to be in good agreement with, among others, the nucleation data of water published by Wood and Walton [131].

Evaluating laboratory data on homogeneous freezing within the framework of CNT, Tanaka and Kimura [52] (Equation (Equation 13)) adopted a linear dependence of the surface tension on temperature corresponding to n=1, which is in between the comparative power values from the literature and the present analysis.

Unlike the temperature dependence of the surface tension, there are only scarce data on its pressure dependence. The simulation data of Espinosa et al. [18] (Figure 1d) revealed a positive pressure coefficient of the surface tension ($\partial \sigma_{\alpha \beta }/\partial p\approx 0.5\;\overset{˚}{A}$ in the range ΔT=(0−50)K). The positive definiteness of the pressure coefficient results in a nucleation rate depression upon increasing pressure, which is used in cryopreservation of biological samples, food, and organs to avoid water freezing and cell damage by application of high pressures (Espinosa et al. [18] (Figure 1d)). The pressure coefficient of the surface tension presented in Table 10 amounts $\partial \sigma_{\alpha \beta }/\partial p\approx 0.06\;\overset{˚}{A}$, which is in qualitative agreement with the simulation data of Espinosa et al. [18] (Figure 1d), even if their value is one order of magnitude larger. However, in view of the completely different approaches underlying the present study and those of Espinosa et al. the agreement is good. Espinosa et al. [18] emphasized that *“the dependence of σ with pressure is totally unknown experimentally. In fact, there is not even a consensus for the experimental value of σ at ambient pressure (there are reported values ranging from 25 to $35\;\mathrm{mJ}\;m^{-2}$* […])”. With reference to the literature Espinosa et al. [18] speculated that $\partial \sigma_{\alpha \beta }/\partial p>0$ originates from pressure-induced breakage of hydrogen bonds in the liquid phase. The diffusion coefficient of water increases with pressure. By hydrogen-bond breaking, the liquid is argued to decrease its structural resemblance to ice and, as the consequence, the surface tension between water and ice increases. We should add, however, that already Jeffery and Austin [31] (Figure 6), giving reference to experimental data from Huang and Bartell [30] for very small droplets (diameter 3nm), presented graphs of the nucleation rate as a function of temperature at isobars p=(0.1,55)MPa, which also reveal a significant decrease of the nucleation rate with increasing pressure. Also the empirical parameterization of the homogeneous nucleation rate of water proposed by Koop et al. [133] predicts a nucleation-rate depression upon increasing pressure (see also Ford [54] (Figure 2)).

Figure 1 displays the normalized ice–water surface tension $\sigma_{\alpha \beta }\left(T,p\right)/\sigma_{\alpha \beta ,m}$ at the coexistence line, i.e., as a function of temperature *T* along the melting pressure curve $p=p_{m}\left(T\right)$ for Equation (Equation 13) according to Jeffery and Austin [31] (Equation (Equation 8)), Equation (Equation 8) according to Schmelzer et al. [109] (Equation (Equation 30)), and Equation (Equation 9) according to Schmelzer et al. [109] (Equation (Equation 32)). Both Equations (Equation 13) and (Equation 8) exhibit the existence of a minimum, which is lost in the linearized form.

The TEOS-10-based limiting values of the Tolman length scale according to Equations (Equation 11) and (Equation 12), respectively, were found to be very close to each other: ${\left.\delta_{\infty }^{\left(T\right)}\right|}_{p=p_{m}^{⋆}}=2.8\;\overset{˚}{A}$ and ${\left.\delta_{\infty }^{\left(p\right)}\right|}_{T=T_{m}^{⋆}}=0.76\;\overset{˚}{A}$. The Tolman length at equilibrium, $\delta_{\infty }$, is independent of the way this equilibrium is approached, i.e., $\delta_{\infty }=\delta_{\infty }^{\left(T\right)}=\delta_{\infty }^{\left(p\right)}$. However, due to several approximations employed in the determination of the generating quantities entering the nonlinear expressions of $\delta_{\infty }$, $\delta_{\infty }^{\left(T\right)}$ and $\delta_{\infty }^{\left(p\right)}$, the equality sign was replaced with the approximate symbol. Apart from that, the determination of $\delta_{\infty }^{\left(T\right)}$ on the hand and $\delta_{\infty }^{\left(p\right)}$ on the other hand is based on different thermophysical properties, the physical uncertainties of which may also contribute to the recognized differences between the isothermal and isobaric paths of the derivation.

Based on the experimentally determined positive temperature coefficient of the surface tension, $\partial \sigma_{\alpha \beta }/\partial T>0$, and previous X-ray diffraction studies indicating an increasingly ice-like structure of liquid water upon increasing undercooling, McDonald [22] (Table 2 and reference therein to Dorsch and Boyd) concluded: *“As the structure of the two phases grow increasingly more similar, it should follow that the surface free energy of the interface between the two phases should decrease towards the zero value that it must exhibit in the limit of complete isomorphism”* (see also Ickes et al. [123]).

Apart from this general conclusion, one can distinguish two cases concerning the behavior of $\sigma_{\alpha \beta }$ at low temperatures:(i)Zeroing the surface tension (but also the thermodynamic driving force) in the T−p plane could be expected by approaching–if it exists–a spinodal of undercooled water. The latter is defined by a line $\left(T_{s},p_{s}\right)$ at which water loses its thermodynamic stability. Based on thermodynamic arguments, the spinodal is defined by zero values of the isodynamic stability coefficients (e.g., Skripov and Baidakov [58], Skripov [57]; Kluge and Neugebauer [134]; Baidakov [61]; Skripov and Faizullin [67]):
(17)$${\left(\frac{\partial T}{\partial {\hat{S}}_{\beta }}\right)}_{p}=\frac{T}{{\hat{c}}_{p,\beta }}=0\;,$$
(18)$$-{\left(\frac{\partial p}{\partial {\hat{V}}_{\beta }}\right)}_{T}=\frac{1}{{\hat{V}}_{\beta }\kappa_{T,\beta }}=0.$$According to Equations (Equation 17) and (Equation 18), the spinodal of undercooled water is approached by ${\hat{c}}_{p,\beta }\to \infty$ and $\kappa_{T,\beta }\to \infty$. At the spinodal, the ice-water surface tension, $\sigma_{\alpha \beta }\left(T,p\right)$ according to Equation (Equation 8), is expected to vanish, as can be deduced from the limiting behavior of the isobaric temperature coefficient of the surface tension:
(19)$${\left(\frac{\partial \sigma_{\alpha \beta }}{\partial T}\right)}_{p}=\frac{\sigma_{\alpha \beta }}{T}+T\frac{\sigma_{\alpha \beta ,m}}{T_{m}\Delta {\hat{S}}_{m}}\left({\hat{c}}_{p,\beta }-{\hat{c}}_{p,\alpha }\right).$$According to Feistel and Wagner [Figure 1) (see also Feistel and Hagen [77,78], Feistel and Wagner [79,80,82], Giauque and Stout [135], and IAPWS R10-06 [83]), the mass-specific heat capacity of ice, ${\hat{c}}_{p,\alpha }$, at atmospheric pressure is a monotonous function of temperature with $\partial {\hat{c}}_{p,\alpha }/\partial T>0$ and
$$\lim_{T\to 0}\frac{{\hat{c}}_{p,\alpha }}{T^{3}}=0.0091\;J\;{\mathrm{kg}}^{-1}\;K^{-4}.$$If a spinodal temperature, $T_{s}$, exists with
$$\lim_{T\to T_{s}}{\hat{c}}_{p,\beta }=\infty \;,$$
one could expect
$$\lim_{T\to T_{s}}{\left(\frac{\partial \sigma_{\alpha \beta }}{\partial T}\right)}_{p}=\infty \;⇝\;\lim_{T\to T_{s}}\sigma_{\alpha \beta }=0.$$(ii)At the Kauzmann temperature $T_{K}$—provided it exists—the temperature dependence of the surface tension is controlled by Equation (Equation 8), i.e.,
$$\sigma_{\alpha \beta }\left(T,p\right)\propto T\Delta S\left(T,p\right).$$By virtue of the definition of the Kauzmann temperature one could expect:
$$\lim_{T\to T_{K}}\Delta S\left(T,p\right)=0\;⇝\;\lim_{T\to T_{K}}\sigma_{\alpha \beta }=0.$$

In a pioneering paper, Skripov and Baidakov [58] provided evidence for the absence of a spinodal in one-component melt crystallization. This study stimulated intensive laboratory and theoretical investigations, and computer simulations on the limits of metastability of undercooled liquids. However, despite enormeous research over many decades there is still much controversy on the existence of a spinodal in undercooled liquids. Our review of the literature disclosed a tendency in the bulk of studies, which supports the proposition of Skripov and Baidakov [58] also for water. Here, we base our consideration on previous studies on the temperature dependence of the isobaric heat capacity, including a van der Waals model, recent computer simulations, and a state-of-the-art EoS for undercooled water. To gain a qualitative picture of the isobaric heat capacity, Gránásy [40] (Figure 2c) adopted a modified van der Waals model proposed by Poole et al. [136], yielding a maximum difference of the isobaric heat capacity between water and ice of $\Delta {\hat{c}}_{p}\;\approx \;{\hat{c}}_{p,\beta }\;-\;{\hat{c}}_{p,\alpha }=5.56\;\mathrm{kJ}\;{\mathrm{kg}}^{-1}\;K^{-1}$ occuring at T=232K. From their MD simulations Moore and Molinero [51] (Figure 1a and references) deduced a maximum isobaric heat capacity of ${\hat{c}}_{p,\beta }\;\approx \;5.56\;\mathrm{kJ}\;{\mathrm{kg}}^{-1}\;K^{-1}$ at the liquid transformation temperature $T_{L}\;\approx \;202\;K$ (defined by the maximum change in density), which is also the maximum change in tetrahedrality and fraction of four-coordinated molecules. Moore and Molinero [51] (see references) noted that $T_{L}$ in their simulations is ≈15K above the singular temperature of the power law, $T_{s}$, derived from a fit of predicted ${\hat{c}}_{p,\beta }$ values using the mW water model of [137], and ≈25K below the $T_{s}\;\approx \;225\;K$ estimated from the experimental values of the heat capacity of water (Speedy and Angell [138], Tombari et al. [139]). In accordance with this, the extrapolation of the new EoS of undercooled water proposed by Holten et al. [85] (Figure 14) into the deeply undercooled range yields a maximum of the isobaric heat capacity of ${\hat{c}}_{p,\beta }\;\approx \;7.5\;\mathrm{kJ}\;{\mathrm{kg}}^{-1}\;K^{-1}$ at T≈228K. The findings of Moore and Molinero [51] and Holten et al. [85] suggest that the temperature coefficient of the surface tension remains finite at $T_{L}$. From Cahn–Hilliard-type density functional calculations for homogeneous ice nucleation in undercooled water Gránásy [40] (Figure 7a) predicted a monotonous behavior of the ice-water surface tension in the temperature interval 160K≤T≤270K with a finite value of $\sigma_{\alpha \beta }\;\approx \;\left(10-15\right)\;\mathrm{mJ}\;m^{-2}$ at T=160K. Hence, there is no resilient empiricism for the accessibility of the state of complete ice-water isomorphism.

### 4.3. Critical Cluster Size

Knowing the thermodynamic driving force for nucleation and the surface tension, the radius of the critical cluster, $R_{\alpha }$, is obtained from Equation (Equation 1). Table 11 contains the values of $R_{\alpha }$ determined using the exact form of the driving force, $\Delta g_{\mathrm{df},c}^{\left(\mathrm{bulk}\right)}=p_{\alpha }-p_{\beta }$ (Equation (Equation 1)) together with $\sigma_{\alpha \beta }\left(T,p\right)≅\sigma_{\alpha \beta ,m}\left[T\Delta \hat{S}\left(T,p\right)\right]/\left[T_{m}\Delta {\hat{S}}_{m}\right]$ according to Equation (Equation 8), and Table 12 shows the corresponding radii determined using the linearized forms of the driving force, ${\left.\Delta g_{\mathrm{df},c}^{\left(\mathrm{bulk}\right)}\left(T,p\right)\right|}_{\mathrm{lin}}$ (Equation (Equation 7)) and the surface tension, $\sigma_{\alpha \beta }\left(T,p\right)$ according to Equation (Equation 9). The critical radius decreases upon decreasing temperature and pressure. For the considered range of ΔT and Δp≤10MPa the radii determined from the different parameter combinations agree quite well, suggesting that the linearization of the driving force and the surface tension captures the temperature and pressure dependencies still very well in this range.

### 4.4. Homogeneous Water-to-Ice Nucleation Rate

To determine the sensitivity of the homogeneous water-to-ice nucleation rate against different formulations of $\sigma_{\alpha \beta }\left(k\right)$ (index k=1,…,3 corresponding to Equations (Equation 13), (Equation 8), and (Equation 9)) and of $\Delta g_{\mathrm{df},c}^{\left(\mathrm{bulk}\right)}\left(l\right)$ (index l=1,…,4 corresponding to Equations (Equation 1), (Equation 5), (Equation 6), (Equation 7)) we employ Equation (Equation 1) for *J* with the kinetic prefactor $J_{\mathrm{kin}}$ taken from Jeffery and Austin [31] (Equation (Equation 1)) (see also Hagen et al. [24] (Equation (Equation 1)); for derivation of $J_{\mathrm{kin}}$ see e.g., Pruppacher and Klett [5] and Hellmuth et al. [13]):(20)$$\begin{array}{ccc} J(k,l) & = & J_{\mathrm{kin}}\left(k\right)\exp\left(-\frac{\Delta G_{c}^{\left(\mathrm{cluster}\right)}\left(k,l\right)}{k_{B}T}\right)\;,\;\\ \Delta G_{c}^{\left(\mathrm{cluster}\right)}\left(k,l\right) & = & \frac{1}{3}A_{\alpha }\left(k,l\right)\sigma_{\alpha \beta }\left(k\right)\;,\;\\ A_{\alpha }\left(k,l\right) & = & 4\pi {\left[R_{\alpha }\left(k,l\right)\right]}^{2}\;,\;R_{\alpha }\left(k,l\right)=\frac{2\sigma_{\alpha \beta }\left(k\right)}{\Delta g_{\mathrm{df},c}^{\left(\mathrm{bulk}\right)}\left(l\right)}.\;\\ J_{\mathrm{kin}}\left(k\right) & = & 2N_{c}\left(\frac{{\hat{\rho }}_{\beta }}{{\hat{\rho }}_{\alpha }}\right)\left(\frac{k_{B}T}{h}\right)\sqrt{\frac{\sigma_{\alpha \beta }\left(k\right)}{k_{B}T}}\exp\left[-\frac{\Delta G_{\mathrm{act}}}{k_{B}T}\right]\;,\;\\ \; & \; & k=1,\ldots 3\;,\;l=1,\ldots ,4. \end{array}$$

The kinetic prefactor represents the diffusive molecular flux across the solid–liquid interface. In Equation (Equation 20), $N_{c}=5.85\times 10^{18}\;m^{-2}$ is the number of monomers of water in contact with unit area of the ice surface, $k_{B}$ is the Boltzmann constant, and *h* the Planck constant. The quantity $\Delta G_{\mathrm{act}}\left(T,p\right)$ denotes the molecular ice-water activation energy. The expression for $\Delta G_{\mathrm{act}}\left(T,p\right)$ used here is based on an empirical Vogel–Fulcher–Tammann (VFT) equation for the self-diffusivity of water (see Jeffery and Austin [31] (Equation (Equation 15) and discussion in Section 5), as well as Appendix B):(21)$$\Delta G_{\mathrm{act}}\left(T,p\right)=k_{B}T\left[\frac{B(p)}{T-T_{⋆}\left(p\right)}-\ln\left(\frac{D_{⋆}\left(p\right)}{D_{0}\left(p\right)}\right)\right].$$

The pressure-dependent self-diffusivity parameters B(p), $T_{⋆}\left(p\right)$, $D_{⋆}\left(p\right)$, and $D_{0}\left(p\right)$ at isobars p=(0.1,10,50,100,150,200)MPa are taken from Jeffery and Austin [31] (Table 2). Table 2 in Jeffery and Austin [31], containing the parameters for the self-diffusivity *D* according to their Equations (11) and (15), is subject of two cumbersome mistakes in the unit annotation. The correct unit assignment in column 2 and 5 of Table 2 must read $D_{⋆/0}\times 10^{8}/m^{2}\;s^{-1}$, and in column 3 the correct annotation is B/K (see e.g., Prielmeier et al. [140] (Table 3); Ludwig [56] (Figure 3a); Hernández de la Peña and Kusalik [48]) (Table II). For details see Appendix B.

Figure 2, Figure 3, Figure 4, Figure 5, Figure 6, Figure 7, Figure 8 and Figure 9 display the nucleation rate $\log_{10}\left[J/\left({\mathrm{cm}}^{-3}s^{-1}\right)\right]$ vs. temperature *T* at isobars p=(0.1,10,50,100,150)MPa. The graph annotation corresponds to the pairwise combinations $\left\{\sigma_{\alpha \beta }\left(k\right),\Delta g_{\mathrm{df},⋆}^{\left(\mathrm{bulk}\right)}\left(l\right)\right\}$ described in Table 13. A common feature exhibited in all figures is a strong increase of the nucleation rate upon decreasing temperature (or increasing undercooling) and decreasing pressure. At atmospheric pressure (Figure 2, Figure 3, Figure 4 and Figure 5) the 12 graphs can be gathered into three group controlled by $\sigma_{\alpha \beta }\left(k\right)$(k=1,…,3). The differences between the nucleation rates caused by the variation in $\Delta g_{\mathrm{df},c}^{\left(\mathrm{bulk}\right)}\left(l\right)$(l=1,…,4) cannot be resolved in Figure 2, i.e., the variation in the driving force does not significantly contribute to the variation in J(k,l). As can be seen from Figure 3, Figure 4 and Figure 5, at atmospheric pressure the differences in the nucleation rate due do variation of the driving force amount less than one order of magnitude. The described grouping of the nucleation rate according to index *k* can also be seen in Figure 6 for p=10MPa. At p=50MPa (see Figure 7) even the differences in the nucleation rates between surface tension indices k=2 and k=3 diminish for the chosen scale of the nucleation rate (differences due to variation of index *l* are not displayed). However, toward the pressure p=100MPa, the differences due to variations in *k* and *l* start to increase (see Figure 8).

As the temperature coefficient of the surface tension (determining the slope of the curve) according to Jeffery and Austin [31] is lower than those for the surface-tension expressions proposed by Schmelzer et al. [109], the surface tension of Jeffery and Austin [31] is larger at lower temperatures, leading to the lowest nucleation rate in Figure 2 (series (k,l)=(1,1−4)). The differences in the nucleation rates between the surface tensions of Jeffery and Austin [31] and Schmelzer et al. [109] are much larger than those between Equation (Equation 8) and Equation (Equation 9) proposed by Schmelzer et al. [109]. This grouping behavior is pronounced at low and moderate pressure (p=(0.1,10)MPa), but starts to diminish at pressures above, i.e., the variation in the nucleation rate becomes more and more controlled by variations in the thermodynamic driving force, which can be seen from the increasing differences between the temperature dependencies of *J* within each of the three groups representing the considered formulations for $\sigma_{\alpha \beta }\left(T,p\right)$ (Figure 9, p=150MPa).

Figure 10 shows the calculated decadic logarithm of the nucleation rate $\log_{10}J$ vs. temperature *T* at atmospheric pressure as in Figure 2 but with the expectation range of the experimental data analyzed by Ickes et al. [122] (Figures 10 and 11). The green-shaded area represents the empirical expectation range of the nucleation rate defined by the scatter among 34 different sources of experimentally derived nucleation rates analyzed by Ickes et al. [122]. The bold green lines represent the envelopes of the scattered data presented in Ickes et al. [122] (Figure 11). The analysis revealed that the empirical temperature coefficient of the nucleation rate, ∂J/∂T, is very well captured by the calculated nucleation rates. However, the expectation range of the calculated nucleation rates due to the variation of the ice–water interfacial energy (red, black, and blue lines) is about a factor 3 larger than the expectation range of the experimental nucleation rates (green lines and green-shaded area). While the employment of the Turbull expression (k=1, blue line, Equation (Equation 13)) leads to a strong underestimation of the nucleation rate, the application of the entropy-based expressions (k=2, red line, Equation (Equation 8), and k=3, black line, Equation (Equation 9)) leads to a strong overestimation of the nucleation rate. This result is a direct consequence of the caculated differences in the ice–water interfacial energy depicted in Figure 1. The difference between the two entropy-based expressions of $\sigma_{\alpha \beta }\left(T\right)$ leads to a calculated expectation range in $\log_{10}J\left(k,l\right)$ which is about as large as the empirical expectation range.

It is interesting to note, that at atmospheric pressure the variation in the thermodynamic driving contributes little to the variation in $\log_{10}J\left(k,l\right)$. The corresponding contribution of the driving force to the calculated expectation range of $\log_{10}J\left(k,l\right)$ remains much smaller than the empirical expectation range. The comparison depicted in Figure 10 confirms that fitted nucleation parameters (such as the Turnbull expression of $\sigma_{\alpha \beta }$) will lose their predictive power when applied independently from the specification of the kinetic pre-factor and thermodynamic driving force underlying the fitting to experimental nucleation rates. The question to be answered is the following: Which physical nucleation parameter is the most conclusive one to move the graph of the calculated nucleation rate into its empirical expectation range defined by the green-shaded area of Figure 10? Ideally, the aim is to remove the degrees of freedom in the nucleation rate calculation by formulation of as much physical constraints as unknown nucleation parameters have to be determined. This question is subject of Part II of the manuscript.

### 4.5. Kauzmann Temperature and Kauzmann Pressure of Water

According to Equation (Equation 14), a positive definiteness of the Kauzmann temperature requires the fulfillment of the inequality $\gamma_{T,m}>1$. For the ice-water system one has $\gamma_{T,m}\;\approx \;1.74$ and $T_{K}=116\;K$ corresponding to $T_{K}/T_{m}^{⋆}\;\approx \;0.42$.

For comparison, Schmelzer et al. [110] (Table 1) reported a ratio of $T_{K}/T_{m}^{⋆}\approx 0.26$ for the glass-forming melt of $2{\mathrm{Na}}_{2}O\cdot 1\mathrm{CaO}\cdot 2{\mathrm{SiO}}_{2}$. The Kauzmann temperature is well below the “no-man’s land” in the water-phase diagram, enclosed between the glass transition (or vitrification) temperature of water, $T_{g}=136\;K$, and the temperature of homogeneous nucleation, $T_{H}\;\approx \;232\;K$ (Moore and Molinero [51]).

Correspondingly, according to Equation (Equation 14) the positive definiteness of the Kauzmann pressure requires the fulfillment of the inequality $\gamma_{p,m}\;>\;0$. For the ice-water system, however, one has $\gamma_{p,m}\;\approx \;-4.7\times 10^{-4}$ originating from $\Delta {\hat{V}}_{m}={\hat{V}}_{\beta }\left(T_{m}^{⋆},p_{m}^{⋆}\right)-{\hat{V}}_{\alpha }\left(T_{m}^{⋆},p_{m}^{⋆}\right)<0$, i.e., at the melting point the mass density of water is higher than that of ice. As a consequence, the Kauzmann pressure attains a negative value of $p_{K}\;=\;-212\;\mathrm{MPa}$ (undercooled liquid under tension). As the pressure has to be decreased in order to initiate crystallization of water, a maximum of the driving force is reconcilable with negative pressure. According to Nada et al. [43] (p. 298), the MD simulations of Matsumoto et al. [41] of ice nucleation and growth in deeply undercooled water revealed nucleation *“only at an extraordinary low negative pressure”*, but did not predict ice nucleation at atmospheric pressure.

Matsumoto et al. [41] emphasized that for systems with a limited number of possible disordered hydrogen-bond network structures, such as confined water, it is relatively easy to locate a pathway from a liquid state to a crystalline structure. In contrast to this, for pure and spatially confined water, MD simulations of freezing were argued to be severely hampered by the large number of possible network configurations that exist. Matsumoto et al. [41] calculated MD trajectories for 512 water molecules for different temperatures, but at T=230K only one out of six trajectories exhibited sucessful crystallization. The authors also performed constant-pressure and constant-temperature trajectory calculations for various sizes of the system, but crystallization was observed only for very small systems (64 molecules). According to Leach [141] (p. 309, Equation (6.13)), the pressure can be determined on the base of MD simulations according to the following relation:$$p=\frac{1}{V}\left[Nk_{B}T-\frac{1}{3}\sum_{i=1}^{N}\sum_{j=i+1}^{N}r_{ij}f_{ij}\right].$$

Here, *V* denotes the volume of the system, *N* the number of molecules, $r_{ij}$ and $f_{ij}$ the distance and the force acting between atomes *i* and *j*. According to this relation, at low number of molecules, *N*, the pressure becomes negativ. Hence, the phrase *“nucleation only at an extraordinary low negative pressure”* employed by Nada et al. [43] (p. 298) refers to a large absolute value of the negative pressure, or equivalently, to a large value of the tensile stress of the nucleating system.

However, it cannot be ruled out that the predicted absence of ice nucleation at atmospheric pressure is affected by uncertainties of current water models (e.g., Ludwig [56], Nada et al. [43], Espinosa et al. [17], Vega and Abascal [45], Hernández de la Peña and Kusalik [48], Vega et al. [49], Moore and Molinero [51]). In any case, the predicted Kauzmann pressure is already below the extrapolated spinodal pressure of water according to the IAPWS-95 formulation (Wagner and Pruß [70] (Figure 7.54)).

In principle, the Kauzmann temperature and pressure could be determined also directly without any approximations by searching for the temperature and pressure at which the equality of the mass-specific entropies and volumes of the both macrophases is fulfilled. This would require an EoS of water, which is valid down to these values of temperature and pressure. The application of TEOS-10, however, is restricted to temperatures equal or higher than the homogenous freezing temperature and to positively definite pressures.

## 5. Summary and Conclusions

Employing the advanced seawater standard TEOS-10, we applied recently developed expressions for the thermodynamic driving force of crystallization and the crystal–melt surface tension to the ice-water system. It was shown that the thermodynamic driving force can be completely determined from thermodynamic properties provided by TEOS-10 for undercooled water and ice. As reference value for the driving force the pressure difference between the ice cluster and the undercooled water was determined. Several approximations of the driving force were evaluated.

The driving force approximation based on linearization of the chemical potentials was demonstrated to deviate by not more than 0.5% from the exact solution in the ranges of temperature and pressure differences 0K≤ΔT≤39K and 0MPa≤Δp≤100MPa. The determination of the driving force by numerical integration of the Gibbs fundamental equation was found to deviate by not more than 0.7% from the exact solution in the ranges 0K≤ΔT≤39K and 0MPa≤Δp≤10MPa. At the Δp=100MPa isobar, the maximum relative deviation exceeded 7% at ΔT=10K. Finally, the determination of the driving force by analytical integration of the linearized Gibbs fundamental equation was found to deviate by not more than 1.8% from the exact solution in the ranges 0K≤ΔT≤39K and 0MPa≤Δp≤10MPa, but at Δp=100MPa the maximum deviation exceeded 50% at ΔT=10K. Fortunately, the high-pressure regions with enhanced error correspond to states with extremely low nucleation rates.

Provided the surface tension at the melting point is given from experiments (serving as an empirical closure parameter), the pressure and temperature dependencies of the surface tension are fully determined from water and ice entropies given by TEOS-10. The linearization of the surface tension was shown to recover the theoretical scaling law in the ranges of temperature and pressure differences 0K≤ΔT≤35K and 0MPa≤Δp≤100MPa with a relative deviation of ≤6%.

Our TEOS-10-based predictions of the nucleation rate revealed pressure-induced deceleration of ice nucleation, which is in qualitative agreement with laboratory experiments and computer simulations. By a special choice of the kinetic prefactor the sensitivity of the nucleation rate against different expressions for the thermodynamic driving force and the surface tensions was analyzed. At atmospheric pressure the variance of the nucleation rate was mainly controlled by the variance in the surface tension. With increasing pressure difference Δp the variance in the nucleation rate was increasingly controlled by the variance in the thermodynamic driving force. The nucleation rate determination is subject to a closure problem, requiring the availability of the surface tension at the melting point and the activation energy. In the case of water, all other thermodynamic quantities are available from TEOS-10. However, owing to the large uncertainties in the activation energy and the melting-point surface tension (as reported in the literature) homogeneous freezing of undercooled water cannot be considered “a work done”.

The temperature and pressure dependencies of the ice-water surface tension follow the le Chatelier–Braun principle, in that the surface tension decreases upon increasing degree of metastability, which favors water freezing and in this way readjustment of the metastable system back to a stable state. The increase of the surface tension with increasing pressure can be explained by the higher thermal expansion coefficient of ice in comparison to water at the melting point. Finally, the calculated values of the Kauzmann temperature and pressure, corresponding to the maxima of the driving force to nucleation, are fully reconcilable with the temperature and pressure dependencies of the driving force and with laboratory findings and computer simulations on the temperature and pressure dependencies of the nucleation rate. The reason for the negative value of the Kauzmann pressure is the higher mass density of water in comparison to that of ice at the melting point.

## 6. Outlook

In Part II of the paper the expressions of the thermodynamic driving force of ice crystallization and the ice–water surface tension derived here will be compared with previously published formulations. In addition, the results of an intercomparison between theoretically predicted and experimentally derived rates of homogeneous ice crystallization will be presented and discussed.

## Figures and Tables

**Figure 1 entropy-22-00050-f001:**
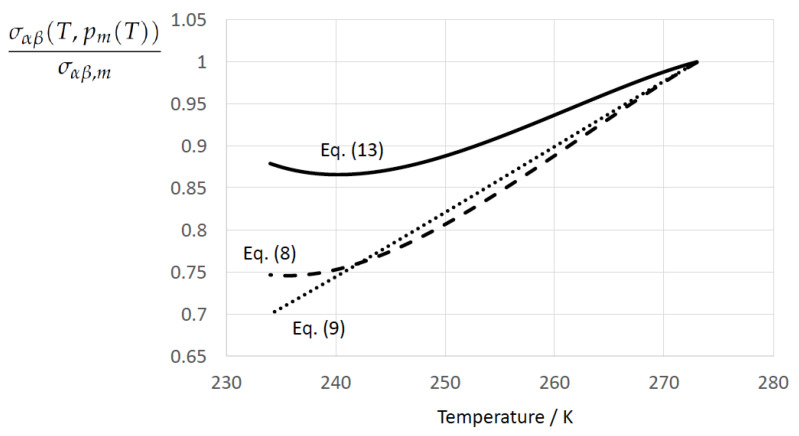
Normalized ice–water surface tension $\sigma_{\alpha \beta }\left(T,p_{m}\left(T\right)\right)/\sigma_{\alpha \beta ,m}$ as a function of temperature T/K along the melting pressure line $p=p_{m}\left(T\right)$. Solid line: Equation (Equation 13) according to Jeffery and Austin [31] (Equation (Equation 8))). Dashed line: Equation (Equation 8) according to Schmelzer et al. [109] (Equation (Equation 30)). Dotted line: Equation (Equation 9) according to Schmelzer et al. [109] (Equation (Equation 32)).

**Figure 2 entropy-22-00050-f002:**
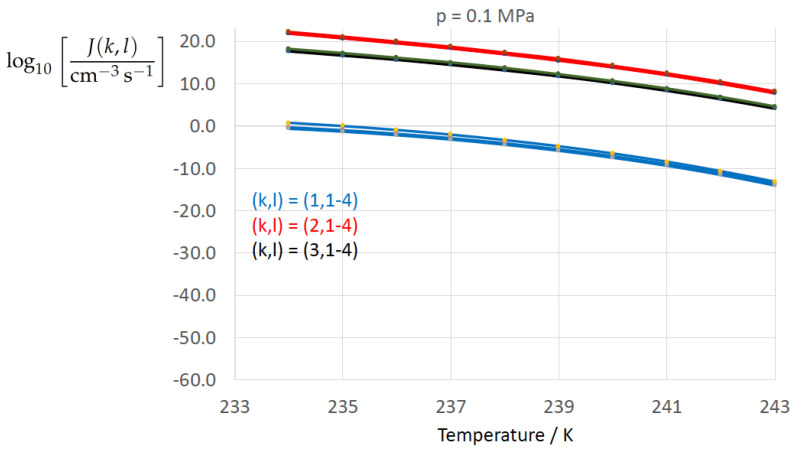
Nucleation rate $\log_{10}\left[J/\left({\mathrm{cm}}^{-3}s^{-1}\right)\right]$ vs. temperature T/K for isobar p=0.1MPa. The graph annotation corresponds to the pairwise combinations $\left\{\sigma_{\alpha \beta }\left(k\right),\Delta g_{\mathrm{df},c}^{\left(\mathrm{bulk}\right)}\left(l\right)\right\}$ described in Table 13.

**Figure 3 entropy-22-00050-f003:**
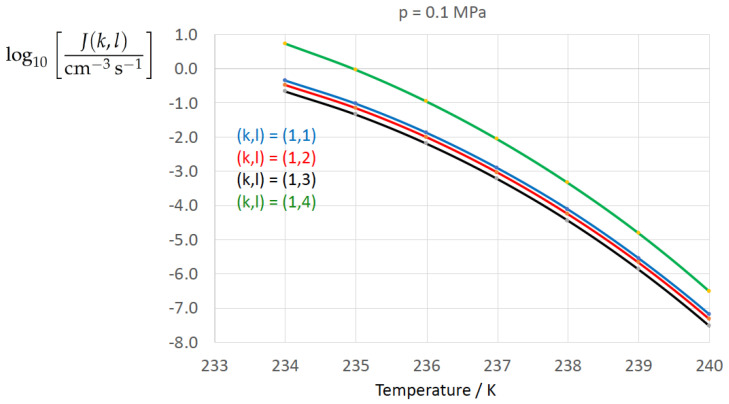
As Figure 2 for k=1 only.

**Figure 4 entropy-22-00050-f004:**
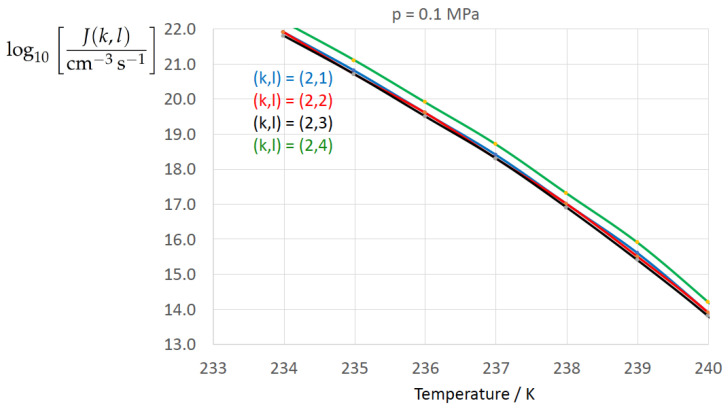
As Figure 2 for k=2 only.

**Figure 5 entropy-22-00050-f005:**
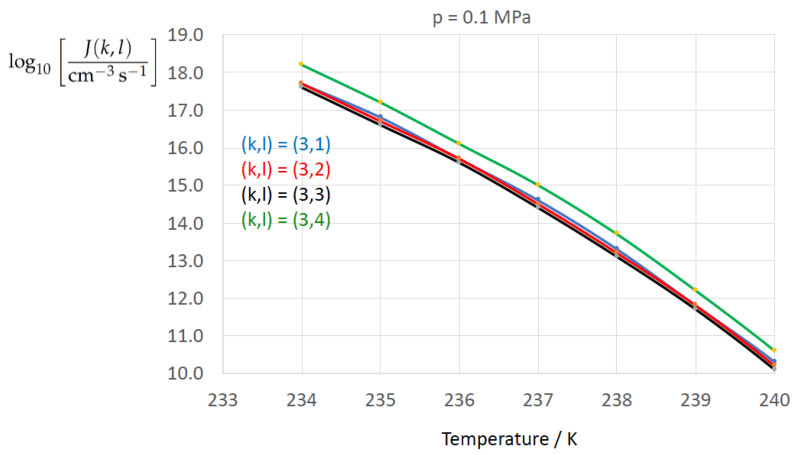
As Figure 2 for k=3 only.

**Figure 6 entropy-22-00050-f006:**
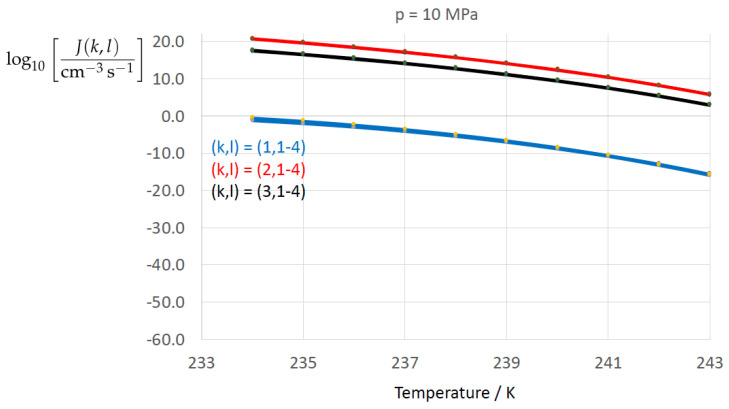
As Figure 2 for isobar p=10MPa.

**Figure 7 entropy-22-00050-f007:**
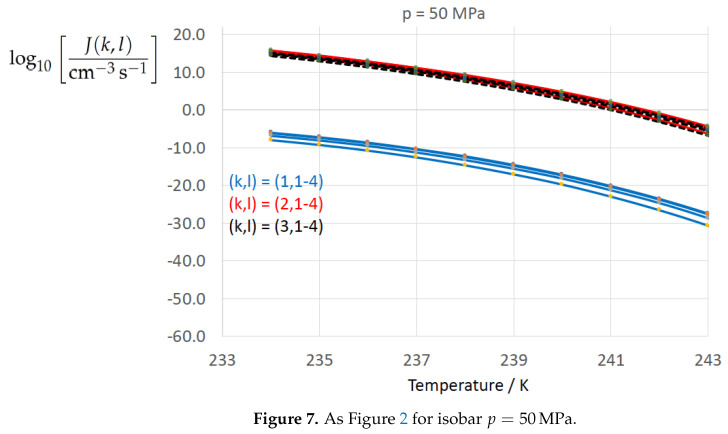
As Figure 2 for isobar p=50MPa.

**Figure 8 entropy-22-00050-f008:**
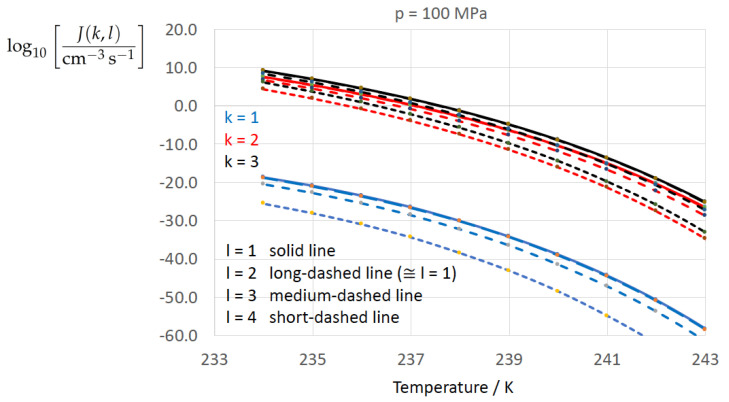
As Figure 2 for isobar p=100MPa.

**Figure 9 entropy-22-00050-f009:**
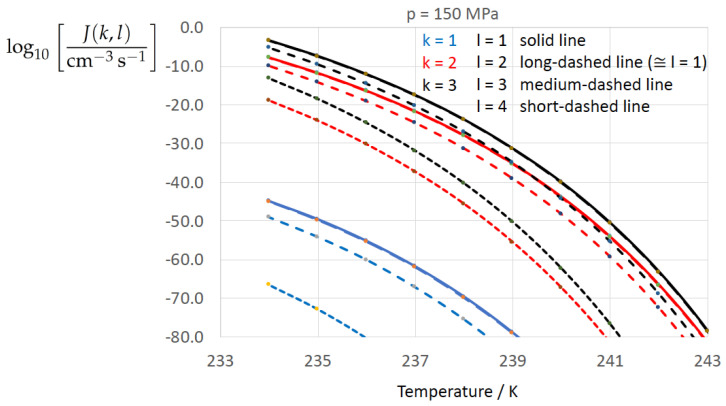
As Figure 2 for isobar p=150MPa.

**Figure 10 entropy-22-00050-f010:**
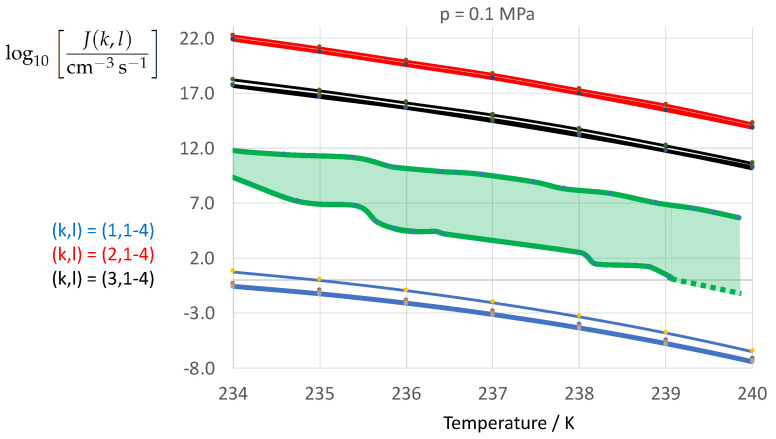
Calculated nucleation rate $\log_{10}\left[J/\left({\mathrm{cm}}^{-3}s^{-1}\right)\right]$ vs. temperature T/K for isobar p=0.1MPa as in Figure 2, but with the expectation range of the experimental data analyzed by Ickes et al. [122] (Figures 10 and 11). The graph annotation corresponds to the pairwise combinations $\left\{\sigma_{\alpha \beta }\left(k\right),\Delta g_{\mathrm{df},c}^{\left(\mathrm{bulk}\right)}\left(l\right)\right\}$ described in Table 13. The green-shaded area represents the scatter of experimental data depicted in Ickes et al. [122] (Figure 11).

**Table 1 entropy-22-00050-t001:** Ratio $\sigma_{\alpha \beta }\left(T,p\right)/\sigma_{\alpha \beta ,m}$ according to Equation (Equation 13) (Jeffery and Austin [31] (Equation (Equation 8))) as a function of undercooling $\Delta T=T_{m}^{⋆}\;-\;T$ and pressure difference $\Delta p=p\;-\;p_{m}^{⋆}$.

ΔT/K	Δp/MPa			
	0	1	10	100
0	1.000	1.000	1.001	1.008
5	0.975	0.975	0.975	0.982
10	0.946	0.946	0.946	0.953
15	0.917	0.917	0.917	0.923
20	0.890	0.890	0.890	0.896
25	0.868	0.868	0.868	0.874
30	0.854	0.854	0.854	0.859
35	0.851	0.851	0.852	0.857
39	0.861	0.862	0.862	0.867

**Table 2 entropy-22-00050-t002:** TEOS-10 SIA library functions used in the present analysis. The SIA equation (last column) refers to the equation number in Wright et al. [88] (Supplement).

Property	Symbol	Unit	FORTRAN Call	SIA Equation
Mass density of water	${\hat{\varrho }}_{\beta }=1/{\hat{V}}_{\beta }$	$\mathrm{kg}\;m^{-3}$	liq_density_si(T,p)	(S11.2)
Mass density of ice	${\hat{\varrho }}_{\alpha }=1/{\hat{V}}_{\alpha }$	$\mathrm{kg}\;m^{-3}$	ice_density_si(T,p)	(S8.3)
Specific Gibbs energy of water	${\hat{G}}_{\beta }$	$J\;{\mathrm{kg}}^{-1}$	liq_gibbs_energy_si(T,p)	(S14.6)
Specific Gibbs energy of ice	${\hat{G}}_{\alpha }$	$J\;{\mathrm{kg}}^{-1}$	ice_chempot_si(T,p)	(S8.1)
Specific enthalpy of water	${\hat{H}}_{\beta }$	$J\;{\mathrm{kg}}^{-1}$	liq_enthalpy_si(T,p)	(14.3)
Specific enthalpy of ice	${\hat{H}}_{\alpha }$	$J\;{\mathrm{kg}}^{-1}$	ice_enthalpy_si(T,p)	(S8.4)
Specific melting enthalpy	$\Delta {\hat{H}}_{M}\left(T\right)$	$J\;{\mathrm{kg}}^{-1}$	temp=set_ice_liq_eq_at_t(T)	
			temp=set_ice_liq_eq_at_p(p)	
			ice_liq_enthalpy_melt_si()	(S23.6)
Specific entropy of water	${\hat{S}}_{\beta }$	$J\;{\mathrm{kg}}^{-1}\;K^{-1}$	liq_entropy_si(T,p)	(S14.4)
Specific entropy of ice	${\hat{S}}_{\alpha }$	$J\;{\mathrm{kg}}^{-1}\;K^{-1}$	ice_entropy_si(T,p)	(S8.5)
Specific isobaric heat capacity of water	${\hat{c}}_{p,\beta }$	$J\;{\mathrm{kg}}^{-1}\;K^{-1}$	liq_cp_si(T,p)	(S14.1)
Specific isobaric heat capacity of ice	${\hat{c}}_{p,\alpha }$	$J\;{\mathrm{kg}}^{-1}\;K^{-1}$	ice_cp_si(T,p)	(S8.2)
Isothermal compressibility of water	$\kappa_{T,\beta }$	${\mathrm{Pa}}^{-1}$	liq_kappa_t_si(T,p)	(S14.9)
Isothermal compressibility of ice	$\kappa_{T,\alpha }$	${\mathrm{Pa}}^{-1}$	ice_kappa_t_si(T,p)	(S8.10)
Thermal expansion coefficient of water	$\alpha_{p,\beta }$	$K^{-1}$	liq_expansion_si(T,p)	(S14.5)
Thermal expansion coefficient of ice	$\alpha_{p,\alpha }$	$K^{-1}$	ice_expansion_si(T,p)	(S8.6)
Melting pressure	$p_{m}$	Pa	ice_liq_meltingpressure_si(T)	(S23.10)
Melting temperature	$T_{m}$	K	ice_liq_meltingtemperature_si(p)	(S23.11)

**Table 3 entropy-22-00050-t003:** TEOS-10-based thermodynamic parameters of the ice-water system at the reference equilibrium state $T_{m}^{⋆}=273.15\;K$ and $p_{m}^{⋆}=0.1\;\mathrm{MPa}$.

Symbol	Equation	Value	Unit
$\Delta {\hat{S}}_{m}$	(Equation 7)	1.221	$\mathrm{kJ}\;{\mathrm{kg}}^{-1}\;K^{-1}$
$\Delta s_{m}$	(Equation 9)	1.119	$\mathrm{MJ}\;m^{-3}\;K^{-1}$
$\Delta {\hat{c}}_{p,m}$	(Equation 7)	2.123	$\mathrm{kJ}\;{\mathrm{kg}}^{-1}\;K^{-1}$
$\Delta {\hat{H}}_{M,m}$	(Equation 7)	333.427	$\mathrm{kJ}\;{\mathrm{kg}}^{-1}$
$\Delta h_{m}$	(Equation 7)	305.659	$\mathrm{MJ}\;m^{-3}$
$\Delta {\hat{V}}_{m}$	(Equation 7)	$-9.069\times 10^{-5}$	$m^{3}\;{\mathrm{kg}}^{-1}$
$\Delta v_{m}$	(Equation 7)	$-8.313\times 10^{-2}$	1
$\Delta \kappa_{T,m}$	(Equation 7)	$3.911\times 10^{-10}$	${\mathrm{Pa}}^{-1}$
$\Delta \alpha_{p,m}$	(Equation 9)	$-2.276\times 10^{-4}$	$K^{-1}$
$\gamma_{T,m}$	(Equation 7)	1.739	1
$\gamma_{p,m}$	(Equation 7)	$-4.704\times 10^{-4}$	1
$\chi_{p,m}$	(Equation 9)	$-2.034\times 10^{-5}$	1
$\delta_{\infty }^{\left(T\right)}$	(Equation 11)	2.8	$\overset{˚}{A}$
$\delta_{\infty }^{\left(p\right)}$	(Equation 12)	0.76	$\overset{˚}{A}$

**Table 4 entropy-22-00050-t004:** Exact thermodynamic driving force of the ice-water system, $\Delta g_{\mathrm{df},c}^{\left(\mathrm{bulk}\right)}=p_{\alpha }-p_{\beta }$ (in units of MPa) according to Equation (Equation 1), as a function of undercooling $\Delta T=T_{m}^{⋆}-T$ and pressure difference $\Delta p=p-p_{m}^{⋆}$.

ΔT/K	Δp/MPa			
	0	1	10	100
0	−0.000	−0.083	−0.849	−9.944
5	5.511	5.429	4.679	−4.333
10	10.847	10.767	10.036	1.130
15	15.996	15.921	15.214	6.443
20	20.948	20.877	20.202	11.602
25	25.687	25.619	24.985	16.605
30	30.187	30.129	29.548	21.456
35	34.419	34.366	33.862	26.158
39	37.563	37.521	37.109	29.820

**Table 5 entropy-22-00050-t005:** Relative deviation of the approximative thermodynamic driving force, ${\left.\Delta g_{\mathrm{df},c}^{\left(\mathrm{bulk}\right)}\right|}_{\mathrm{approx}}$ according to Equation (Equation 5), from the exact driving force, $\Delta g_{\mathrm{df},c}^{\left(\mathrm{bulk}\right)}$ according to Equation (Equation 1), i.e., $\left[{\left.\Delta g_{\mathrm{df},c}^{\left(\mathrm{bulk}\right)}\right|}_{\mathrm{approx}}-\Delta g_{\mathrm{df},c}^{\left(\mathrm{bulk}\right)}\right]/\Delta g_{\mathrm{df},c}^{\left(\mathrm{bulk}\right)}$ in percent, as a function of undercooling $\Delta T=T_{m}^{⋆}-T$ and pressure difference $\Delta p=p-p_{m}^{⋆}$.

ΔT/K	Δp/MPa			
	0	1	10	100
0	−	−	−	−
5	−0.029	−0.028	−0.026	−
10	−0.062	−0.062	−0.054	−0.005
15	−0.087	−0.095	−0.083	−0.031
20	−0.115	−0.119	−0.116	−0.064
25	−0.143	−0.141	−0.138	−0.085
30	−0.164	−0.172	−0.165	−0.115
35	−0.195	−0.191	−0.182	−0.133
39	−0.206	−0.202	−0.207	−0.151

**Table 6 entropy-22-00050-t006:** Relative deviation of the numerically determined thermodynamic driving force on the base of the Gibbs fundamental equation, ${\left.\Delta g_{\mathrm{df},c}^{\left(\mathrm{bulk}\right)}\right|}_{\mathrm{num}}$ according to Equation (Equation 6), from the exact driving force, $\Delta g_{\mathrm{df},c}^{\left(\mathrm{bulk}\right)}$ according to Equation (Equation 1), i.e., $\left[{\left.\Delta g_{\mathrm{df},c}^{\left(\mathrm{bulk}\right)}\right|}_{\mathrm{num}}-\Delta g_{\mathrm{df},c}^{\left(\mathrm{bulk}\right)}\right]/\Delta g_{\mathrm{df},c}^{\left(\mathrm{bulk}\right)}$ in percent, as a function of undercooling $\Delta T=T_{m}^{⋆}-T$ and pressure difference $\Delta p=p-p_{m}^{⋆}$.

ΔT/K	Δp/MPa			
	0	1	10	100
0	−	−	−	−
5	−0.068	−0.080	−0.199	−
10	−0.141	−0.153	−0.260	−7.063
15	−0.205	−0.225	−0.325	−2.331
20	−0.272	−0.288	−0.394	−1.937
25	−0.338	−0.348	−0.453	−1.814
30	−0.398	−0.417	−0.516	−1.777
35	−0.466	−0.474	−0.570	−1.764
39	−0.509	−0.516	−0.624	−1.766

**Table 7 entropy-22-00050-t007:** Relative deviation of the analytically determined thermodynamic driving force on the base of the linearized Gibbs fundamental equation, ${\left.\Delta g_{\mathrm{df},c}^{\left(\mathrm{bulk}\right)}\right|}_{\mathrm{lin}}$ according to Equation (Equation 7), from the exact driving force, $\Delta g_{\mathrm{df},c}^{\left(\mathrm{bulk}\right)}$ according to Equation (Equation 1), i.e., $\left[{\left.\Delta g_{\mathrm{df},c}^{\left(\mathrm{bulk}\right)}\right|}_{\mathrm{lin}}-\Delta g_{\mathrm{df},c}^{\left(\mathrm{bulk}\right)}\right]/\Delta g_{\mathrm{df},c}^{\left(\mathrm{bulk}\right)}$ in percent, as a function of undercooling $\Delta T=T_{m}^{⋆}-T$ and pressure difference $\Delta p=p-p_{m}^{⋆}$.

ΔT/K	Δp/MPa			
	0	1	10	100
0	−	−	−	−
5	−0.084	−0.119	−0.504	−
10	−0.117	−0.157	−0.530	−49.992
15	−0.079	−0.132	−0.534	−11.294
20	0.033	−0.023	−0.484	−7.888
25	0.242	0.183	−0.348	−6.774
30	0.587	0.506	−0.118	−6.342
35	1.111	1.025	0.263	−6.211
39	7.758	1.649	0.710	−6.254

**Table 8 entropy-22-00050-t008:** Relative deviation (in percent) of the ratio $\sigma_{\alpha \beta }\left(T,p\right)/\sigma_{\alpha \beta ,m}$ according to Equation (Equation 8) (Schmelzer et al. [109] (Equation (Equation 30))) from the reference ratio given by Equation (Equation 13) (Jeffery and Austin [31] (Equation (Equation 8))) as a function of undercooling $\Delta T=T_{m}^{⋆}-T$ and pressure difference $\Delta p=p-p_{m}^{⋆}$.

ΔT/K	Δp/MPa			
	0	1	10	100
0	0.000	0.012	0.104	−0.112
5	−2.551	−2.531	−2.367	−2.134
10	−4.923	−4.892	−4.638	−3.866
15	−7.477	−7.432	−7.061	−5.619
20	−10.520	−10.456	−9.928	−7.629
25	−14.399	−14.309	−13.561	−10.134
30	−19.547	−19.418	−18.349	−13.386
35	−26.502	−26.314	−24.768	−17.632
39	−34.191	−33.802	−31.440	−21.883

**Table 9 entropy-22-00050-t009:** Relative deviation (in percent) of the ratio $\sigma_{\alpha \beta }\left(T,p\right)/\sigma_{\alpha \beta ,m}$ according to Equation (Equation 9) (Schmelzer et al. [109] (Equation (Equation 32))) from the reference ratio given by Equation (Equation 13) (Jeffery and Austin [31] (Equation (Equation 8))) as a function of undercooling $\Delta T=T_{m}^{⋆}-T$ and pressure difference $\Delta p=p-p_{m}^{⋆}$.

ΔT/K	Δp/MPa			
	0	1	10	100
0	0.000	0.012	0.125	1.258
5	−2.478	−2.465	−2.348	−1.170
10	−4.615	−4.601	−4.479	−3.251
15	−6.736	−6.722	−6.595	−5.314
20	−9.099	−9.084	−8.952	−7.622
25	−11.969	−11.954	−11.817	−10.445
30	−15.642	−15.626	−15.487	−14.086
35	−20.417	−20.402	−20.261	−18.853
39	−25.202	−25.186	−25.047	−23.657

**Table 10 entropy-22-00050-t010:** Temperature and pressure coefficients of the surface tension, $\partial \sigma_{\alpha \beta }/\partial T$ and $\partial \sigma_{\alpha \beta }/\partial p$ according to Equation (Equation 15), as functions of undercooling $\Delta T=T_{m}^{⋆}-T$ and pressure difference $\Delta p=p-p_{m}^{⋆}$.

ΔT/K	$\left(\partial \sigma_{\alpha \beta }/\partial T\right)/\left(10^{-4}\;J\;m^{-2}\;K^{-1}\right)$					$\left(\partial \sigma_{\alpha \beta }/\partial p\right)/\left(10^{-2}\;\overset{˚}{A}\right)$
	at $p=p_{m}\left(T\right)$	Δp/MPa				
		0	1	10	100	
0	3.133	3.133	3.134	3.144	3.238	6.354
5	2.93	2.872	2.873	2.881	2.97	6.238
10	2.731	2.631	2.632	2.64	2.722	6.122
15	2.541	2.409	2.409	2.417	2.494	6.005
20	2.361	2.204	2.205	2.212	2.284	5.889
25	2.191	2.016	2.017	2.024	2.091	5.773
30	2.032	1.844	1.845	1.851	1.914	5.657
35	1.884	1.686	1.686	1.692	1.751	5.540
39	1.773	1.568	1.569	1.574	1.630	5.447

**Table 11 entropy-22-00050-t011:** Critical radius, $R_{\alpha }=2\sigma_{\alpha \beta }/\Delta g_{\mathrm{df},c}^{\left(\mathrm{bulk}\right)}$ (in units of nm) according to Equation (Equation 1), using the exact form of the driving force, $\Delta g_{\mathrm{df},c}^{\left(\mathrm{bulk}\right)}=p_{\alpha }-p_{\beta }$ according to Equation (Equation 1), and the surface tension, $\sigma_{\alpha \beta }\left(T,p\right)≅\sigma_{\alpha \beta ,m}\left[T\Delta \hat{S}\left(T,p\right)\right]/\left[T_{m}\Delta {\hat{S}}_{m}\right]$ according to Equation (Equation 8), as a function of undercooling $\Delta T=T_{m}^{⋆}-T$ and pressure difference $\Delta p=p-p_{m}^{⋆}$.

ΔT/K	Δp/MPa			
	0	1	10	100
0.0	−	−	−	−
5	10.771	10.936	12.720	−
10	5.181	5.221	5.620	50.643
15	3.313	3.331	3.502	8.451
20	2.375	2.385	2.481	4.458
25	1.807	1.814	1.877	2.955
30	1.422	1.427	1.475	2.168
35	1.136	1.141	1.183	1.686
39	0.943	0.950	0.995	1.419

**Table 12 entropy-22-00050-t012:** Critical radius, $R_{\alpha }=2\sigma_{\alpha \beta }/\Delta g_{\mathrm{df},c}^{\left(\mathrm{bulk}\right)}$ (in units of nm) according to Equation (Equation 1), using the linearized forms of the driving force, ${\left.\Delta g_{\mathrm{df},c}^{\left(\mathrm{bulk}\right)}\left(T,p\right)\right|}_{\mathrm{lin}}$ according to Equation (Equation 7), and of the surface tension, σ(T,p) according to Equation (Equation 9), as a function of undercooling $\Delta T=T_{m}^{⋆}-T$ and pressure difference $\Delta p=p-p_{m}^{⋆}$.

ΔT/K	Δp/MPa			
	0	1	10	100
0.0	−	−	−	−
5	10.788	10.956	12.787	−
10	5.204	5.245	5.659	101.918
15	3.342	3.361	3.538	9.558
20	2.412	2.422	2.520	4.840
25	1.854	1.860	1.922	3.158
30	1.482	1.487	1.529	2.296
35	1.217	1.220	1.251	1.771
39	1.054	1.056	1.080	1.480

**Table 13 entropy-22-00050-t013:** Indexing of the nucleation rate J(k,l) for three different formulations of the surface tension $\sigma_{\alpha \beta }\left(k\right)$ (k=1,…,3) and four different formulations for the thermodynamic driving force $\Delta g_{\mathrm{df},c}^{\left(\mathrm{bulk}\right)}\left(l\right)$ (l=1,…,4). The number in each table cell is the number of the graph in Figure 2, Figure 3, Figure 4, Figure 5, Figure 6, Figure 7, Figure 8 and Figure 9.

$\sigma_{\alpha \beta }\left(k\right)$		$\Delta g_{\mathrm{df},c}^{\left(\mathrm{bulk}\right)}\left(l\right)$			
		l=1	l=2	l=3	l=4
		**Equation (Equation 1)**	**(Equation 5)**	**(Equation 6)**	**(Equation 7)**
k=1	Equation (Equation 13)	1	2	3	4
k=2	Equation (Equation 8)	5	6	7	8
k=3	Equation (Equation 9)	9	10	11	12

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
