# Peer review of "Ice-Crystal Nucleation in Water: Thermodynamic Driving Force and Surface Tension. Part I: Theoretical Foundation"

_entropy, 2019, doi:10.3390/e22010050_

Round 1

Reviewer 1 Report

The authors present a very detailed work on quantifying key quantities used in predicting the crystal nucleation rate of water based on Classical Nucleation Theory and the TEOS-10 equation of state for water and ice.  The results should be of interest to the community, and the manuscript should be published.

One of the appealing characteristics of the manuscript is the level of detail and review of concepts along with extensive references. The extensive appendices provide a review of many aspects of supercooled water, and therefore the manuscript would provide useful reference reading.  However, since the appendices are not necessarily crucial to the results, and given their extent, I did not check them thoroughly.

Given the interest in water at negative pressure, I regret that the EOS used is only for positive pressure.  Perhaps the authors could discuss how they might extend their work to negative p.

I only have minor suggestions for improvement. 

My most major suggestion is to  include axes labels for all graphs.

Otherwise:

line 45 "others”

Check wording on line 121

It would be helpful to include in the Introduction something about the Kauzmann temperature and pressure

Eq 10 page Please define l_infty (and Eq. A.50 on page 33)

Line 215, Ibidem refers to Kauzmann’s work, or Debenedetti’s?

Line 216 glas -> glass

line 231 should start with “and” before “referred” I think.  The sentence looks like it should be corrected in some way at least.

p7, regarding Eq. 11 an 12 for delta_inf^T and delta_inf^P, could the authors please explain to the reader whether these two quantities are expected to have the same values?  Certainly in table 2 they are different.  Comments on this would be appreciated.

line 279, regarding Delta g_{df, c}^bulk = p_alpha - p_beta, according to Eq. 1.  Just to confirm, p_alpha is determined by solving equation 3, yes?  I think it would be beneficial to reiterate this here.

line 281, is the use of the word “undercooled” appropriate in the case where the the driving force is negative?  The water is not undercooled at these conditions.

line 284.  I don’t understand “i.e. starting at p^*_m, the pressure difference must be Delta p = p - p^*_m < 0  to crystallize water.”  While the driving force decreases for increasing p, water certainly crystallizes for Delta p > 0. All values in the table are for Delta p >= 0.  Perhaps add the words “at T^*_m”.   

between line 303 and 304: “as function” -> “as a function”

For the captions of Figs. 2-6, I think it would be useful for the reader if the identity of the curves from bottom to top could be given.  Eg, for Fig. 2:  Series labels for curves from bottom to top are 3, 2, 1, 4 (in a close band), 5, 6/12? (close together), 7, 8 (close together),…  Actually, I can not distinguish 12 curves. If some of them overlap, it would be useful to point that out.  This information is outlined in the text, but it would be very useful to have in the captions.

line 458, last word: “low” or “large”?

Author Response

Please find attached our reply.

Olaf Hellmuth

Reviewer 2 Report

A theoretical analysis of ice nucleation in supercooled water is
developed. Based on bulk thermodynamic properties of ice and liquid water
the authors propose a temperature/pressure dependence of both the driving
force for ice nucleation and the ice-water interfacial free energy. Whit these,
the authors provide estimates of the ice nucleation rate and the Kauzmann state point.
This is a novel interesting analysis and deserves publication in Entropy after
the authors take into account the following issues:

First of all I would like to point out that this paper has been submitted as an "article".
In many sections of the lengthy appendix it looks more like a review paper or even a master thesis.
I am not aware of the policy of the journal, but if this paper is not expected to be a review, I
would prompt the authors to remove those sections of the Appendix that simply review previous work
and leave only those that are new and necessary to support the results of the main part of the paper,
which are in fact novel and interesting.

Regarding the main paper, my main criticism is that no attempt has been made to compare with
experimental data of the predicted nucleation rate. A figure with experimental/simulation
data of the nucleation rate alongside the predictions made by the authors should be provided for 1 bar, where
there is plenty of published data in the papers the authors refer to in their work. By comparing
to experimental data, the authors will be able to discriminate and discuss which of their 12 formulations
for the nucleation rate yields more accurate results.

Then, for the most successful formulation a graph where the nucleation rate for 1 bar is compared to
that of higher pressures should be also included. If possible, comparison with experimental/simulation
data should also be included in the figure. This graph will show the decelerating effect of pressure on
ice nucleation at a glance.

A comparison with experimental/simulation data should be also attempted for sigma.
Using the reference value of 31.2 mJ/m^2 mentioned in the paper,
the authors should make a plot showing the variation of sigma with temperature at 1 bar for the
sigma equation that better describes the nucleation rate (8, 9 or 14) and compare the obtained curve
with other proposals for sigma(T) at 1 bar given in the Ickes paper or in simulation papers.

In Fig. 1 the authors say that they report sigma along the coexistence pressure line. However, it seems
to me that they are reporting sigma at the reference coexistence pressure (1 bar). Please clarify this point
and, if needed, correct the description of the figure in the caption and in the main text.

Alongside Fig. 1, an analogous figure showing the variation of sigma with pressure at constant
temperature should be provided. It would be very interesting to show in a figure the pressure dependence of sigma.

The x and y axis of all figures should be labelled. It is not enough to describe the axis in the caption.

Why are the authors using a sea water EoS to describe pure water? Is not there a more accurate EoS devoted only
to pure water? Is this the best available EoS?

Is sigma_{alpha,beta,m} in Eq. 8 different from that in Eq. 9? If not, the definition given below Eq. 9 should
be moved below Eq. 8. If they are different, that in Eq. 9 should be distinguished from that in Eq.

Why is Eq. 14, that describes an alternative way of getting the sigma dependence with p and T, is presented in
the Results section and not in Section 2.3?

It is not clear what the authors conclude regarding the ice-water spinodal. Do they predict its existence
with their thermodynamic framework? Please clarify this in the discussion.

In table 10, why is capital "M" used as a subscript?

Remove the labels "Datenreihen" from the figures. Write all labels in English.

Author Response

Please find attached our response.

Olaf Hellmuth

Round 2

Reviewer 2 Report

In view of my first report, where I proposed to include a comparison with
experimental and simulation data, the authors suggest to split their work
in two papers, leaving the comparison for the second one because they
consider it requires too much new work. In fact, if they perform the
comparison as they suggest, a lot of bibliographic analysis work is
required because they propose to compare, one by one, all factors that
contribute to the final nucleation rate. However, the comparison I ask for is
much simpler: just including a few experimental data in a
log(J)(T) figure like Fig. 2 or 3 would be enough for me to accept the paper.
These data appear, for instance, in Figs. 10 or 11 of Ref. 124.
(Ickes et al PCCP 2015). I am just asking to include experimental data
of J alongside the parametrizations made by the authors in order to compare both
and discuss which is the best parameterization.

The other comments I made have been properly addressed.

Author Response

please see the attachment below
